# An approach to the permeation mechanism of learning transfer and teaching strategy in physical education based on complex network

Xin Feng[1,2], Jiapei Li[2], Shuhui Hu[2], Yi Zhao[3,4]*, Long Chen[4], Nan Wang[5]

**1** School of Economics and Management, Yanshan University, Qinhuangdao, China, **2** The School of Management, Hebei GEO University, Shijiazhuang, China, **3** The School of Economics, FuDan University, Shanghai, China, **4** The College of Physical Education & Health, East China Normal University, Shanghai, China, **5** The School of Economics and Management, Hebei University of Technology, Tianjin, China

* zhao_yi@fudan.edu.cn

## Abstract

Learning transfer is widely present in the learning of all kinds of knowledge, skills and social norms, and is one of the important phenomena of learning, and the reasonable use of transfer is conducive to improving the learning effect of students and the quality of teaching. This study starts from the data of college students' academic performance, takes real students' academic performance as a sample, measures the relevance of courses through students' academic performance, constructs various networks of learning transfer, and studies the topology and evolution of the networks to clarify the essential laws of learning transfer and put forward suggestions for the optimization of teaching strategies. Finally, using complex network analysis to analyze and mine the data on college students' academic performance, the article quantifies the overall structure of the courses and their hidden connections in a global and dynamic manner, and discovers the inheritance relationship between the courses, the clustering characteristics and the basic pattern of learning transfer. It also provides a platform for exploring the differences in the course structure of different majors and the learning transfer of male and female students.

## 1. Introduction

Learning transfer refers to the influence of learning results obtained in the activity of learning some materials on other learning. It is a complex and systematic phenomenon, which has existed widely in the learning of knowledge, skills and behavior norms. Through the study of grammar learning, Zhou et al. found that transfer has both positive and negative effects. To use these effects to the greatest extent, the learners as well as teachers should use as much positive transfer as possible as the most important phenomena of learning [1]. Learning transfer can be divided into positive transfer and negative transfer. Positive transfer, also known as "facilitating transfer", refers to that one kind of learning plays a positive role in promoting

**Data Availability Statement:** There are restrictions on publicly sharing the data used in this study, which were obtained from https://portal2020.ecnu.edu.cn/mydashboard. Only the teacher has access

to this website to obtain the results of the students he/she teaches. In these data, there is some students' personal information, including student ID, gender, etc. Therefore, The College of Physical Education & Health, East China Normal University does not recommend that it be publicly visited. Although we have hidden some of the privacy information in the process of research, there is still some that may involve the privacy of students. However, other researchers may send data access requests to Zhongyin Zhang, (zyzhang@tyxx.ecnu. edu.cn).

**Funding:** This study received funding from the following grants: National Natural Science Foundation of China grant No.11905042, awarded to XF; Youth Fund for Humanities and Social Sciences Research for the Ministry of Education grants No.16YJC630022 (awarded to XF) and 20YJC870005 (awarded to JL); Funded Project of "333 Talent Project" in Hebei Province, grant No. A202001015, awarded to XF; and Youth Project of Humanities and Social Sciences Research for the Colleges and Universities of Hebei Province, grant No.SQ201111, awarded to YZ.

**Competing interests:** The authors have declared that no competing interests exist.

another kind of learning, while negative transfer plays an obstacle role. In the learning process, flexible use of positive transfer can achieve twice the result with half the effort. The study of learning transfer helps to explore the essence and laws of learning to better construct a learning theory system. The study of learning transfer is not only theoretically significant but also considerably practically significant, which helps to guide the teaching process and improve teaching efficiency. Firstly, traditional studies are mostly based on basic learning transfer theory assumptions, and they used educational statistics-related methods to perform correlation analysis and regression analysis on various problems in the transfer process. No doubt this can provide some meaningful references, however, it can only provide microscopic and partial course information and transfer behavior. It cannot comprehensively consider learning transfer from the perspective of global, dynamic, and learning behavior complexity. In addition, the relationship between the emerging big data, the overall structure of the course, and students is often impossible to consider, let alone analyze the rules and abilities of students' learning transfer. While the complex network is exactly an interdisciplinary aspect that studies the structural characteristics and dynamic evolution of complex systems. The network structure composed of intricate relationships between notes on a certain time scale is used to explore its structural characteristics and evolution laws. The relationship between things is the objective basis of learning transfer [2], and the edge learning transfer in a complex network is an objective reflection of the internal relationship in essence, which is quantifiable and computable. It is possible to model and study learning transfer from a system level from the perspective of dynamic quantification, which reflects not only the scientificity and superiority of complex networks for learning transfer research, but also the essence of learning transfer networks to some extent. Therefore, this study's interpretation of the permeation mechanism of learning transfer at different network levels will be very helpful for understanding the learning transfer process. It can effectively link relevant objects and apply them in the teaching process, further exploring the essence and laws of transfer and greatly improving the effectiveness of teaching.

## 2. Literature review

As a vital part of higher education, learning transfer plays a decisive role whether from the perspective of the design of the teaching system in higher education or the perspective of the cultivation of students' learning abilities. Therefore, it has also aroused close attention from relevant scholars. In recent years, research on learning transfer has mainly focused on the application of transfer theory in mathematics teaching and how to improve transferability. Also for the research of mathematics, Xiaoqin Li [3] proposed specific application strategies of learning transfer theory in mathematics teaching, while Qingfeng Huang proposed teaching strategies to cultivate and improve the ability of mathematics learning transfer based on the affecting factors of transfer ability [4]. Similar to Qingfeng Huang, aiming at the influencing factors of transferability, Quixia Huang [5] proposed relevant strategies from three aspects: knowledge and skills, processes and methods, and emotional attitudes and values.

Zhongtang Zhao [6] and others teased out the knowledge points of C language under the guidance of the learning transfer theory. And a large number of scholars also have focused their researches on how to improve transfer capabilities. The virtual environment provides good conditions for implementing "learning by doing", Goldin and others used games to train 6-year-old children's executive ability, which affects children's academic performance [7]. Mettler and Pinto used games as the carrier of knowledge transfer to realize the transfer of engineering management knowledge [8]. Trinchero and other scholars divided 931 elementary school students into two experimental groups and a control group [9]. Chess training improved the mathematical ability of the experimental group students, and verification

heuristic teaching could promote learning transferability. Overseas studies on the use of information literacy games to improve the ability of learning transfer are more in-depth and present the characteristics of cross-disciplinary integration. In the related research on language transfer, the academic community has begun to realize the complexity of language transfer as a dependent variable [10]. The study of transfer is no longer confined to language ontology, but has gradually turned to research fields such as cognition and concept transfer, subsequently, some researches on cross-disciplinary fusion have emerged. Pengfei Lei and others probe into the phenomenon of language transfer in classroom foreign language learning environment based on dynamic system theory [11]. Skill transfer is also a major branch of learning transfer. In the teaching and training of sports disciplines, motor skill transfer is often used, and the academic circles increasingly attach importance to it. Xuan Wang probed into the precautions of using motor skill transfer starting from analyzing the forms of motor transfer [12]. Guang-mao Wang took the transfer of sports skills of the separated net as the research object, implemented literature, logical reasoning and other methods to expound and analyzed the phenomenon of transfers, then it demonstrates theoretical guidance on how to play the role of positive transfer for physical education and training [13]. The main ideas for the study of learning transfer between various types of disciplines teaching mentioned above are mostly the use of theoretical models for reasonable logical reasoning supplemented by the use of questionnaires or comparative experiments among students, but there are some problems with such research:

Most of the research objects are samples from one class or several classes, and the randomness of small samples may cause the status of reaction to be incomplete. More importantly, it focuses on the transfer between a single learning element or a single subject, so it has a strong purpose and pertinence, which is conducive to the design of the technical strategy of the experiment, but the scope of the experimental object is too small, and it can only be of reference value in a single field. It lacks a systematic network framework and cannot complete the natural extension from a single subject to multiple subjects. Many frontier teachers explore the design of the teaching plan on learning transfer from the perspective of teachers. To some extent, it can effectively guide the coursework, but it lacks examining the inherent quantification of learning transfer from the perspective of students.

Most of researches above are only the logical deduction of theoretical models whether it is the application of transfer theory or the study of improving transferability, The more in-depth is the use of educational statistics to perform some descriptive analysis and multivariate statistical analysis, it can provide a certain reference for curriculum classification and learning transfer, but it cannot reflect the relevance of the curriculum and the complexity of the transfer from the overall network structure. The information diffusion and penetration between different network nodes is regarded as learning transfer from the perspective of complex networks, studying the essential properties of learning transfer from the topological structure of the complex network, these are undoubtedly more global but can not provide us with a deeper connection of the transfer phenomenon. Therefore, it is urgent to find a method that can explore the essence and laws of transfer phenomenon, so as to better grasp and apply the laws of transfer. Additionally, the theory and methods of complex networks, especially multiple complex networks, can exactly meet the needs of research.

Since Watts and Barabas et al. proposed small-world networks and BA scale-free networks, the complex network theory has entered a new chapter [14, 15]. Since then, it was discovered that a large number of complex systems in the real world can be described through complex networks. From the perspective of complex networks, studying various complex systems and examining the interactions between individuals reveals the evolution and internal laws of complex systems. The research on the subject of complex networks is also constantly developing.

From the topological characteristics of single-layer networks to the compound effects of multiple networks, new research directions have emerged in complex networks. Henrique F. de Arruda et al. [16, 17] used complex networks for text classification, supervised classification, and network models describing local topology and dynamic characteristics of function words were used to distinguish information from imaginary documents. And in 2017, when they studied the use of complex network knowledge acquisition, a multi-agent random walk model was constructed. It was pointed out that most of the dynamic parameters had little impact on the knowledge acquisition process. In the local range, the selection of control dynamic parameters had little impact on the performance of the considered knowledge network. Massimo Stella et al. [18] found the importance of integrating multi-relational word interaction in a psycholinguistic framework by using complex network research.

S. Gómez and other scholars explored the linear diffusion between multiple networks and constructed the Laplacian matrix to obtain the time coefficient of linear diffusion of a multi-layer complex network [19]. A. Solé-Ribalta made a profound study on the role of Laplacian matrix in multiple complex networks based on the research of S. Gómez [20]. There is a large amount of literature using complex networks to study complex systems in the real world, and it is not uncommon in the field of education and teaching. Q Zou [21] and others proposed a construction method of teaching evaluation network model based on complex network theory. Gaowei Yan [22] and others took the courses, knowledge units, and knowledge points of automation specialty as the nodes, and built the directed knowledge network of automation specialty with their prior learning relationship as the edge [20]. They combined the characteristics of mining with instructional design and learning guidance, and innovated teaching methods to improve students' learning ability, which can have a positive impact on the undergraduate teaching of automation specialty.

The construction of a knowledge network or course model by using complex network theory is scientific and operable, which has certain extensibility and applicability. By studying the complex network characteristics of teaching arrangements, the node degree distribution of teaching courses reflects the close connection degree between courses. Through the arrangement of teaching courses based on complex networks, it can help to improve the quality of the automatic arrangement of teaching course networks. The learning transfer behavior has the same characteristics as permeation and diffusion in complex networks. On the one hand, permeation means a subtle process. From the perspective of education, the impact on different professional courses is chronically diffusional. On the other hand, permeation expresses a two-way process, for example, a has an impact on B, B has an impact on C, and they interact and interweave with each other. The permeation of learning transfer behavior helps to improve students' understanding of the curriculum. Permeation in learning transfer is not only an action process of communication, but also reflects the permeation thought of learning transfer.

To some extent, it is more intuitive to understand the essence of learning transfer to study the transfer using the model produced by the multiple complex networks.

## 3. Empirical analysis

### 3.1 Data acquisition and processing

This paper uses the Jupyter tool under Python to crawl all the students' grades (from the academic year of 2007–2008 to the academic year of 2019–2020) under the public database of East China Normal University, and obtains items 78337 of data, including student ID, student name, student gender, major, course, course type, year and student score. Through the data screening of this score table, the two majors with the largest number of students are selected: sports training and physical education. To unify the year, the students' grades from the 2009–

2010 academic year through the 2019–2020 academic year are selected for both majors. We use Python again to sort the professional basic compulsory courses of the two majors in descending order according to the number of students in the courses, and select the top 8 professional basic compulsory courses with the largest number of students in the two professional courses, of which 7 are the same. After sorting out, the students' academic records of the two majors are obtained, that is, each row represents the performance of each student, and each column represents the score of each student in each course. After eliminating the sample of students whose course scores do not exist, we obtain 218 sample sizes of students' scores of sports training and 244 sample sizes of physical education majors.

## 3.2 Basic statistical analysis

A basic statistical analysis on seven identical courses of the two majors is performed, and the average result is shown in the table below. The comparative average score of 7 courses in two majors is shown in **Table 1** and the comparative line chart of two majors is shown in **Fig 1**.

The comparison chart of the average score of different courses shows that the overall score distribution of seven courses in the two majors is similar, and the variation trend of different courses is similar. In order to understand the specific distribution of course scores, the normal distribution test is carried out for the distribution of course scores. Take Sports Introduction and Sports Sociology as examples. As is shown in **Fig 2**.

The results of the normal distribution test of Sports Introduction course scores show that the results show a quasi-normal distribution, and the scores in the middle section are basically in normal distribution, but the high segment and the low segment have obvious fluctuation. Mean score of Sports Introduction for sports training major $m_1$ is about 75.5551, and the standard deviation $s_1$ is about 6.2194. However, the mean score of Sports Introduction for sports education major $m_2$ is about 78.3033 and the standard deviation $s_2$ is about 5.9724. The operating results show that the scores distribution of sports training majors is more intensive, and the different values between the maximum and minimum are smaller. The scores distribution of Sports Sociology also shows such characteristics. As is shown in **Fig 3**.

The mean score of Sports Sociology for sports training majors $m_1$ is about 70.3257, the standard deviation $s_1$ is about 12.7069. Meanwhile, the mean score of physical education $m_2$ is about 77.5615, and the standard deviation $s_2$ is about 77.5615. The mean score comparison graph intuitively shows that the mean score of sports education major is higher than that of sports training major, and the normal distribution test of each course also shows that the score distribution interval of sports education major is smaller.

There are two possible reasons for the performance distribution: first, the overall situation of students majoring in sports education is better than that of students majoring in sports training; Second, the professional nature of sports training determines the importance of basic knowledge, which leads to teachers' stricter requirements for students majoring in sports training. The Sports Sociology also has obvious fluctuation in the low section and the high

**Table 1. Average score of 7 courses in two majors.**

| Courses / Majors | Sports Physiology | Sports Anatomy | Sports Statistics | Sports Psychology | Sports Introduction | Sports Sociology | Sports Research Methods |
|---|---|---|---|---|---|---|---|
| Sports training | 67.73 | 68.89 | 67.65 | 66.12 | 75.56 | 70.33 | 77.65 |
| Physical education | 74.33 | 77.75 | 72.17 | 72.00 | 78.30 | 77.56 | 79.41 |

Tips: Two decimal places are kept for the results.

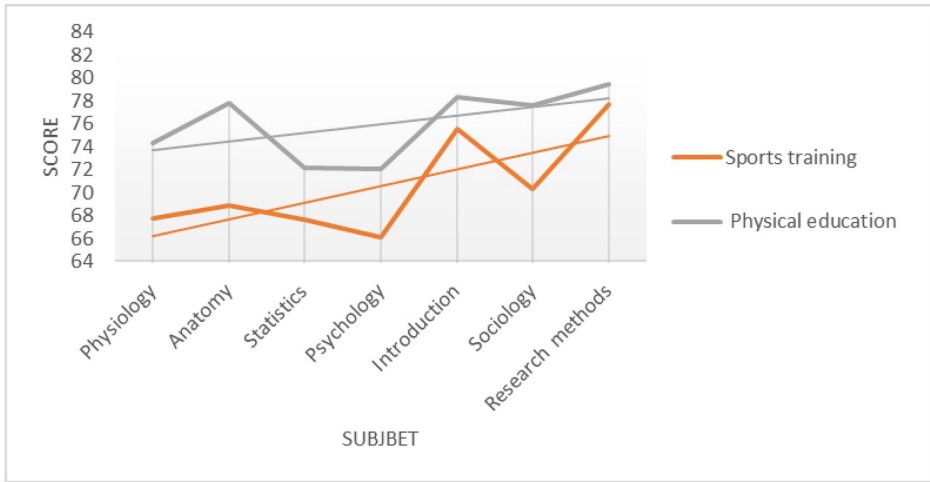

**Fig 1. The comparative line chart of two majors.**

section, due to the fact that the sample amount of the score is not large enough to allow for a certain deviation.

Conducting statistics on the variance of the seven courses shown in **Table 2**, the results also show that the samples deviation is large. The transfer analysis through the mean value cannot reflect the individual characteristics of each student and cannot obtain a systematic curriculum relationship.

At present, most of the empirical tests of learning transfer training take one or two classes as samples for teaching, and then use the mean score and variance of the experimental group and control group for p-value test. The sample size is so small that the randomness is strong, which may lead to the failure to fully reflect the current situation. Secondly, there is a lack of a coping mechanism for the changes caused by individual differences. Therefore, in the fourth part of this paper, the learning transfer mechanism of sports-education teaching process is discussed by constructing a single and multi-layer complex network.

## 4. Modeling

In this section, multiple complex networks were applied to study the permeation mechanism of learning transfer. And the large-scale and multi-scale real academic achievements over the

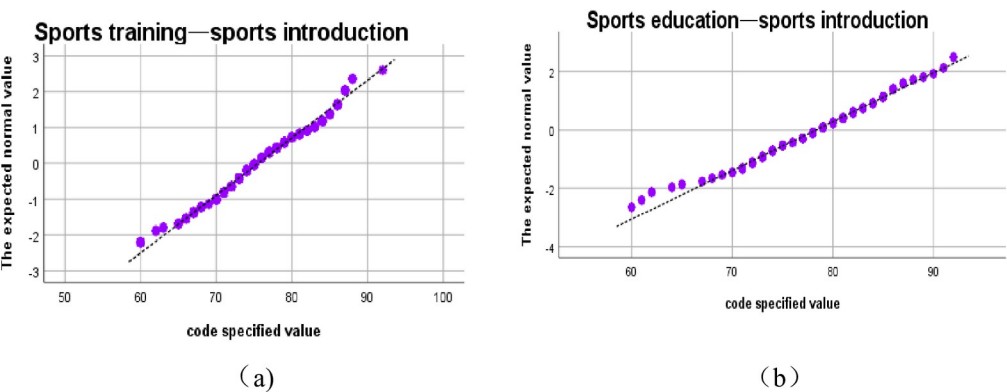

(a)                                                     (b)

**Fig 2. Normal distribution test of Sports Introduction courses.**

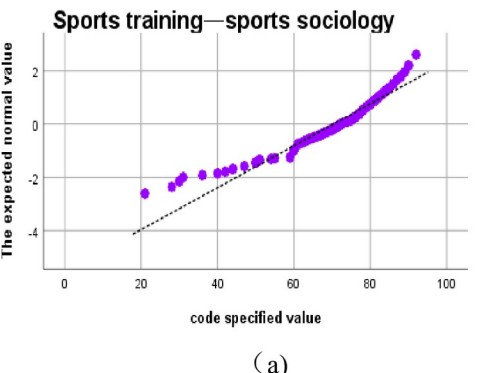 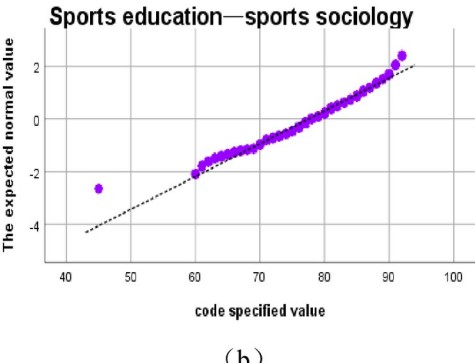

（a） （b）

**Fig 3. Normal distribution test of Sports Sociology.**

years were used as samples to construct the complex network of course and the permeation and transmission mechanism within and between the learning transfer network was discussed. Therefore, the research content of this part is the single-layer and multi-complex network of learning transfer. The basic research ideas and methods are to measure the relevance of courses through students' academic scores, to build a complex network. The information presented in the network connection diagram is used to calculate the network parameters of the complex network which is used to describe the permeation mechanism and generalization nature of learning transfer, and then to observe the permeation propagation law of learning transfer in the course network.

Firstly, the learning transfer process of a single major's internal study was established, and the learning transfer network of intra-specialty was constructed to describe the permeation mechanism of learning transfer among different courses within the specialty.

Secondly, based on clarifying the permeation mechanism of learning transfer within the specialty, the multiple network linear diffusion formula, namely the time-continuity equation of linear diffusion in a multiple systems consisting of M network layers (Formula 1), is used to study the multiple complex network permeation and diffusion mechanism of learning transfer among multiple majors.

$$\frac{dx_i^2}{d_t} = D_\alpha \sum_{j=1}^{N} w_{ij}^\alpha (x_j^a - x_i^a) + \sum_{\beta=1}^{M} D_{\alpha\beta}(x_i^\beta - x_i^\beta) \tag{1}$$

Where $x_j^a$ represents the state of node $j$ in layer $\alpha$. The first term indicates the transfer rate between nodes of different courses in the same layer. The diffusion coefficient $D_\alpha$ within the layer and the inter-layer diffusion coefficient $D_{\alpha,\beta}$ represents different diffusion capacities within and between layers, which are generally different values.

**Table 2. The variance of seven courses in two majors.**

| Courses / Majors | Sports Physiology | Sports Anatomy | Sports Statistics | Sports Psychology | Sports Introduction | Sports Sociology | Sports Research Methods |
|---|---|---|---|---|---|---|---|
| Sports training | 127.66 | 136.37 | 153.60 | 118.02 | 38.68 | 161.46 | 100.05 |
| Sports education | 80.78 | 102.24 | 137.39 | 91.93 | 35.67 | 64.53 | 75.53 |

Tips: Two decimal places are kept for the results.

The second term of the formula indicates the learning transfer rate between nodes of the same course in different layers, so as to scientifically quantify the basic law of intra-specialty and inter-specialty learning transfer. The inter-layer diffusion coefficient represents the diffusion ability between different layers in a multi-layer complex network. It can quantitatively and specifically reflect the diffusion ability between different layers. The main reason for setting the interlayer diffusion coefficient is the influence between different specialties and the influence of different disciplines within the specialty. The strength of these two influences is different. Even if the same course in the different professional teaching process, its emphasis and role are not the same. Through the interlayer diffusion coefficient, this paper studies the interaction of different majors learning the same course. Aiming at the problem that the influence between layers and within layers is different, it is of practical significance to put forward the method of layered analysis.

Compared with the method of logical deduction based on a theoretical model, which is used by most studies at this stage, the multi-complex network can be used to explore the diffusion mechanism of learning transfer from a more systematic and comprehensive perspective, to realize the natural extension and deep expansion of talent training from single discipline to multiple disciplines.

Then, the student-centered teaching design was conducted under the guidance of the complex network permeation mechanism of learning transfer, to formulate more scientific and reasonable teaching strategies and talent enhancement programs to promote the all-round development of learners' learning ability.

## 4.1 Learning transfer in a monolayer complex network

Firstly, the real score data were used as the sample, and the rank correlation coefficient of the score was used to describe the correlation between different courses. And the degree of curriculum correlation was used to construct the boundary relation of the curriculum network. Let K represent the threshold value of correlation. If the correlation of courses is greater than K, there would be a boundary relation between courses. Otherwise, it is not connected. By setting different thresholds, the rich correlation between different courses is displayed dynamically.

Based on the successful construction of the network, this paper explored the complicated relationship between courses by combining the methods of calculating the quantitative index of the complex network and the qualitative analysis of the professional training program.

**4.1.1 The single-layer network construction.**   The score data is a score table of 462 students and 7courses. Each row represents different courses' scores of the same student, and each column represents different students' scores of the same course. A student's learning ability is constant. The learning ability of a course can be directly reflected in the score of the course. The correlation degree of different course scores of the same students, can reflect what the correlation between courses of the student is. Through the correlation of multiple student samples to different course variables, the similarity between different course variables can be obtained. The rank correlation coefficient is used to measure the linear correlation between courses. Then, the quantitative index value is obtained, which is the basis of constructing the course network.

In addition, the construction of a single-layer network is completed under certain threshold conditions. The purpose of setting a threshold is to make single-layer network and multi-layer network have an appropriate scale. In the measurement of correlation, the threshold is similar to the classification tree in statistical analysis. In the process of edge joining, the edge connection and node degree have a certain scale, neither too much nor too little, to observe the relationship between courses.

After the correlation of 7 basic courses of sports major in East China Normal University being calculated, different network diagrams of course links are obtained by dynamically changing the threshold K, as is shown in **Fig 4**.

Among them, the threshold value of Fig 4A is 0.3, the threshold value of Fig 4B is 0.4, and the threshold value of Fig 4C is 0.5. (when the threshold value is 0.2, the graph shows that the link points are almost the same as the threshold value of 0.3, and when the threshold value is 0.6, there are few links between courses.) When the threshold K is changed to increase by 0.1, the number of effective edges in the network can be observed to decrease significantly. It is found that if there are too few edges, a large number of isolated nodes will be formed, which will lose the significance of establishing a network; if there are too many connected edges, it will form a fully connected network, and there is no way to describe the characteristics of the network between nodes. Therefore, these two values are not conducive to the study of the problem from the perspective of the network. At the same time, the two courses in the network graph have effective edges, which indicates that the two courses are closely related under the threshold condition. In the network, if a course has more courses related to it, it means that the node where the course is located is the key node in the network. The more degree a course has, it is an important node in the network, that is, the more important the course is in the whole course network. To sum up, the network link graph with a threshold of 0.5 is selected to better highlight the status of core courses.

**4.1.2 The single-layer network analysis.** *1) The node degree.* The degree of a node refers to the number of edges connected to that node. Obviously, the larger the node degree in the network, the more important the node is. The height node defined in this paper is calculated by comparing the node degree under the same threshold. When the threshold value is higher, the number of effectively connected edges is reduced and the node degree fluctuation of different courses is obvious, the node with higher node degree is divided. Therefore, we need to select the appropriate threshold for node degree analysis and classification. According to the statistics of the node degree of the single-layer network shown in **Table 3**, the node core degree of Sports Anatomy, Sports Psychology and Sports Physiology in the network belongs to the

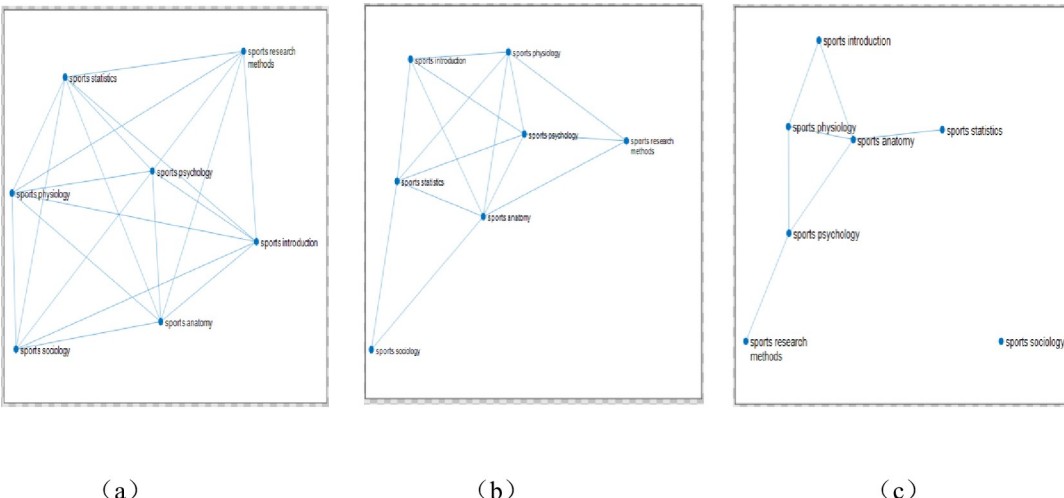

（a）                      （b）                      （c）

**Fig 4. The linked network of 7 courses of sports major from 2009 to 2019.** (Fig 4 shows that after calculating the correlation of seven professional basic courses of 462 students majoring in sports training and physical education in East China Normal University, the different course linked network diagrams are got through dynamically changing the threshold K. The seven courses are respectively Sports Physiology, Sports Anatomy, Sports Statistics, Sports Psychology, Sports Introduction, Sports Sociology, Sports Sociology. The threshold value of figure a is 0.3, that of figure b is 0.4, and that of figure c is 0.5.).

**Table 3. The node degree statistics.**

| Courses | Sports Physiology | Sports Anatomy | Sports Statistics | Sports Psychology | Sports Introduction | Sports Sociology | Sports Research Methods |
|---|---|---|---|---|---|---|---|
| Node degree | 3 | 4 | 1 | 3 | 2 | 0 | 1 |
| Node degree | 5 | 6 | 5 | 5 | 4 | 2 | 3 |
| Node degree | 6 | 6 | 6 | 6 | 6 | 5 | 5 |

first echelon, followed by Sports Introduction, Sports Statistics, Sports Research Methods, and Sports Sociology.

*2) The shortest path length and node betweenness.* The network nodes' shortest path length which is constructed under the condition of a threshold value of 0.4 is as shown in **Table 4**. The shortest path length between nodes could intuitively delineate the close degree between any two courses, and reflect the ability to transfer between different courses. Combining the network diagram, we could find the shortest path between two nodes. On this basis, the optimal migration path between the two courses is established. And through the shortest path length between nodes, the number of the shortest path which passes any nodes can be calculated. Then, the node betweenness is computed, which is the quantitative characterization of the transition function of nodes in the network. If the betweenness of the node is bigger, the shortest path between any two nodes is more. The node betweenness of the seven courses is evaluated as shown in **Table 5**:

Because the sample size of this study is 462, the amount of data is small, and the fluctuation of students' score data is small, to make the results more precise and accurate, the shortest path length between any two courses is reserved to four decimal places. It improves accuracy, which plays an important role in the derivation of experimental conclusions.

The results show that most differences of the shortest path length and average shortest path length between courses are not big, but six groups of courses, "Sports Sociology" and "Sports Physiology", "Sports Sociology" and "Sports Psychology", "Sports Sociology" and "Sports Introduction", "Sports Sociology" and "Sports Research Methods", "Sports Research Methods" and, "Sports Statistics", "Sports Research Methods" and "Sports Introduction" have longer shortest paths. And it is also consistent with the core degree ranking of the courses through node betweenness analysis. The combination of path longer courses comes from the courses at the intermediate or later core level.

*3) Clustering coefficient.* The clustering coefficient expresses the extent to which nodes are adjacent to each other. That is "the possibility that your two friends are also friends for each other". The average clustering coefficient of all nodes is called the average clustering coefficient C or the clustering coefficient of the whole network. Moreover, the sample size of this paper is

**Table 4. The shortest path length for any two courses (2 decimal places are reserved).**

| | Sports Physiology | Sports Anatomy | Sports Statistics | Sports Psychology | Sports Introduction | Sports Sociology | Sports Research Methods |
|---|---|---|---|---|---|---|---|
| Sports Physiology | 0 | 0.65 | 0.49 | 0.63 | 0.56 | 0.89 | 0.48 |
| Sports Anatomy | 0.65 | 0 | 0.55 | 0.54 | 0.60 | 0.47 | 0.46 |
| Sports Statistics | 0.49 | 0.55 | 0 | 0.47 | 0.47 | 0.40 | 0.97 |
| Sports Psychology | 0.63 | 0.54 | 0.47 | 0 | 0.49 | 0.88 | 0.53 |
| Sports Introduction | 0.56 | 0.60 | 0.47 | 0.49 | 0 | 0.87 | 1.01 |
| Sports Sociology | 0.89 | 0.47 | 0.40 | 0.88 | 0.87 | 0 | 0.93 |
| Sports Research Methods | 0.48 | 0.46 | 0.97 | 0.53 | 1.01 | 0.93 | 0 |

**Table 5. The betweenness of each course node.**

| | Sports Physiology | Sports Anatomy | Sports Statistics | Sports Psychology | Sports Introduction | Sports Sociology | Sports Research Methods |
|---|---|---|---|---|---|---|---|
| The node betweenness | 1 | 1 | 3 | 1 | 0 | 0 | 0 |

not large, clustering is easy and the results are intuitive, reflecting the real situation. The analysis of node degree, shortest path length and node betweenness in single layer network is proved. The clustering results show that each discipline and other disciplines have clustering phenomena, but the degree is different, which paves the way for the following study of complex network models.

The clustering coefficient of network nodes under the threshold value of 0.4 is shown in **Table 6**. The clustering coefficient of the whole network C is about 0.46. If C is 0, it means that every node in the network is isolated. If C equals to 1, it means that the network is global coupled. That is to say, any two nodes are connected.

In addition to the above analysis, to increase the integrity and reliability of the experiment, we also divide the data set into two non-overlapping years, and build the corresponding network spanning rom 2009 to 2019. The analysis is as follows: after processing the data set, considering the different courses taken by students in each academic year, the sample size between each group of years and the problem that the number of courses will decrease with the number of years grouped, the years can be divided into two groups: 2009–2016 and 2017–2019. The courses taken by students in the two groups are Sports Physiology, Sports Anatomy, Sports Statistics, Introduction to Psychology, Sports, Sports Sociology, and Sports Scientific Research Method. By observing the course link network of the two groups of years, and comparing the node degree, shortest path length and node betweenness in different years groups, it is helpful to find the core courses in the research samples more intuitively. The course link network diagram of each stage and the corresponding specific analysis are as follows:

Firstly, from 2009 to 2016, the total sample size of the selected students is 396, and the course link network diagram is shown in **Fig 5**.

When the threshold value increases from 0.4 (Fig 5A) to 0.5 (Fig 5B), a significant decrease in the number of effective edges can be observed. When the threshold value is 0.4, the correlation between courses is relatively close, so the threshold value of 0.4 is selected for analysis. The degree of Sports Anatomy course is the most, which is an important node in the network. It proves the importance of the course in the course network. It will be analyzed from four aspects: node degree statistics, shortest path length, betweenness of each course node and clustering coefficient of each course node, as shown in **Table 7**.

The degree of a node refers to the number of edges connected to the node. The larger the node is, the more important it is. In 2009–2016, when the threshold is 0.4, the node degree of Sports Physiology and Sports Anatomy is the largest, so these two disciplines occupy the core position in the network. It is also verified in the analysis of the shortest path length between any two courses, as shown in **Table 8**.

**Table 6. The clustering coefficient of each course node (2 decimal places are reserved).**

| | Sports Physiology | Sports Anatomy | Sports Statistics | Sports Psychology | Sports Introduction | Sports Sociology | Sports Research Methods |
|---|---|---|---|---|---|---|---|
| The clustering coefficient | 0.41 | 0.30 | 0.39 | 0.43 | 0.56 | 0.55 | 0.61 |

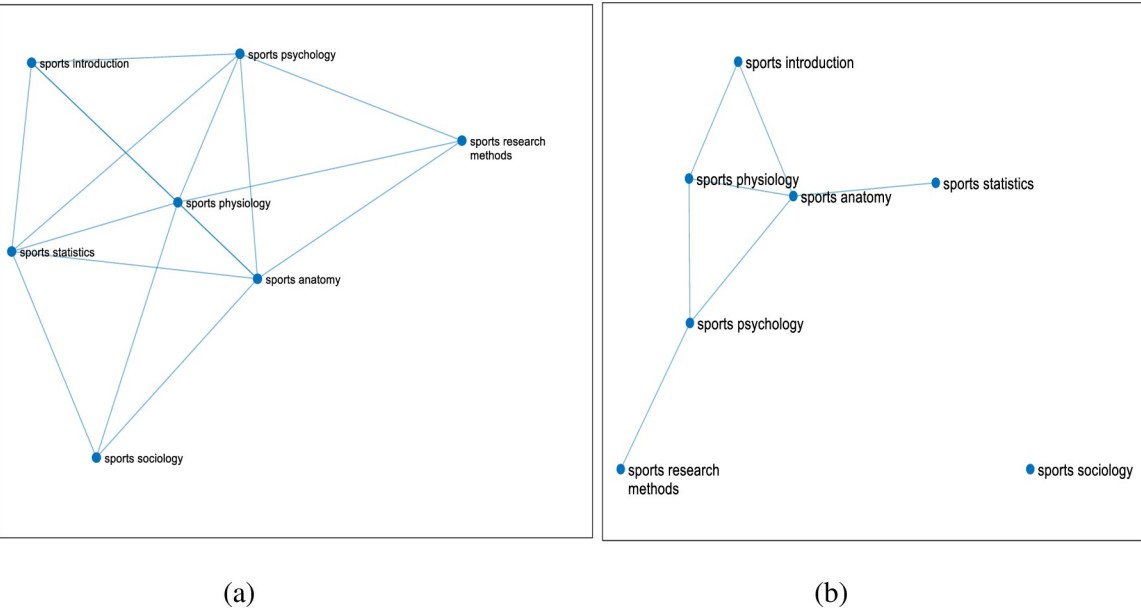

(a) (b)

**Fig 5. The linked network of 7 courses of sports major from 2009 to 2016.** (Fig 5 shows that after calculating the correlation of seven professional basic courses of 396 students majoring in sports training and physical education from 2009 to 2016 in East China Normal University, the different course linked network diagrams are got through dynamically changing the threshold K. The seven courses are respectively Sports Physiology, Sports Anatomy, Sports Statistics, Sports Psychology, Sports Introduction, Sports Sociology, Sports Sociology. The threshold value of figure (a) is 0.4, and the threshold value of figure (b) is 0.5.).

The shortest path length of network nodes constructed under the condition of the threshold value of 0.4 is shown in Table 8. The calculation results show that the average shortest path length De≈0.60. It can be seen in Table 8 that the shortest path length between Sports Physiology and the other four courses is between 0.4 and 0.56 except for Sports Anatomy and Sports Psychology courses; except for Sports Physiology courses, Sports Anatomy and the other five courses. The shortest path length of the course is between 0.4 and 0.59; the shortest path length of Sports Statistics and other five courses is between 0.4 and 0.51 except for Sports Research Method course; the shortest path length between Sports Psychology and other four courses is between 0.4 and 0.54 except Sports Physiology and Sports Sociology; except Sports Sociology and Sports Scientific Research Method course, the shortest path length of Sports Introduction and the other four courses is between 0.4 and 0.59; the length range of the above shortest paths is less than the average shortest path length. From the above analysis, we can see that Sports Anatomy and Sports Statistics are closely related to other courses. Students should pay special attention to these two courses in the learning process, and teachers should also focus on observing and cultivating students' learning of these two courses in the teaching process.

According to the betweenness of the node, the transition function of the node in the network is quantitatively described. The higher the betweenness of the node, the higher the core degree of the course. As can be seen from **Table 9**, the node betweenness value of Sports Physiology is 2, and that of Sports Statistics is 2, which is larger than that of other courses. Therefore,

**Table 7. The node degree statistics.**

| Courses | Sports Physiology | Sports Anatomy | Sports Statistics | Sports Psychology | Sports Introduction | Sports Sociology | Sports Research Methods |
|---|---|---|---|---|---|---|---|
| Node degree | 3 | 4 | 1 | 3 | 2 | 0 | 1 |
| Node degree | 6 | 6 | 5 | 5 | 4 | 3 | 3 |

**Table 8. The shortest path length for any two courses (2 decimal places are reserved).**

|  | Sports Physiology | Sports Anatomy | Sports Statistics | Sports Psychology | Sports Introduction | Sports Sociology | Sports Research Methods |
|---|---|---|---|---|---|---|---|
| Sports Physiology | 0 | 0.65 | 0.44 | 0.62 | 0.56 | 0.40 | 0.47 |
| Sports Anatomy | 0.65 | 0 | 0.51 | 0.54 | 0.59 | 0.48 | 0.45 |
| Sports Statistics | 0.44 | 0.51 | 0 | 0.44 | 0.46 | 0.43 | 0.91 |
| Sports Psychology | 0.62 | 0.54 | 0.44 | 0 | 0.49 | 0.87 | 0.50 |
| Sports Introduction | 0.56 | 0.59 | 0.46 | 0.49 | 0 | 0.89 | 0.99 |
| Sports Sociology | 0.40 | 0.48 | 0.43 | 0.87 | 0.89 | 0 | 0.87 |
| Sports Research Methods | 0.47 | 0.45 | 0.91 | 0.50 | 0.99 | 0.87 | 0 |

it is considered that Sports Physiology and Sports Statistics have a high degree of intermediary, play a role of communication bridge, and play a related role in the ability transfer of the curriculum, which should be paid attention to in the curriculum design and assessment.

As shown in **Table 10**, by calculating the clustering coefficient of each course node, it is found that the clustering coefficient of the whole network is C≈0.45, The node clustering coefficient of Sports Research Methods course is 0.60, which is the highest among the seven courses. It indicates that this course has the highest correlation coefficient with other courses, and it is closely combined with other courses, and belongs to the upper-level course in the course group; the node clustering coefficient of Sports Introduction and Sports Sociology is 0.53, second only to Sports Research Methods, so these two courses are closely combined with other courses and belong to basic courses in the curriculum group. Through the analysis, we can see that the three courses with higher clustering coefficient (Sports Research Methods, Sports Introduction and Sports Sociology) are closely related to other courses in the seven professional courses, but their status and levels are different.

Secondly, 2017–2019 is selected as the second stage, and the curriculum link network diagram is constructed as shown in **Fig 6**.

According to Fig 6, whether the threshold value is 0.4 or the threshold value is 0.5, the degree of Sports Anatomy in the network is the most, and it is an important node in the network, that is, the course is the most important in the course network. In the following, we will analyze from four aspects: node degree statistics, the shortest path length, the betweenness of each course node and the clustering coefficient of each course node. The node degree statistics in **Table 11** are as follows:

According to Table 11, the node degree of Sports Physiology and Sports Anatomy is significantly higher than that of the other five disciplines. It can be preliminarily considered that Sports Physiology and Sports Anatomy are at the core of teaching and learning. Moreover, it is also verified in the analysis of the shortest path length between any two courses in the following table (Table 12).

As shown in **Table 12**, the shortest path length of network nodes constructed under the condition of a threshold value of 0.4 is calculated, and the calculation results show that the average shortest path length De≈0.88. It can be seen in Table 12 that the shortest path length

**Table 9. The betweenness of each course node.**

|  | Sports Physiology | Sports Anatomy | Sports Statistics | Sports Psychology | Sports Introduction | Sports Sociology | Sports Research Methods |
|---|---|---|---|---|---|---|---|
| The node betweenness | 2 | 0 | 2 | 1 | 0 | 0 | 0 |

**Table 10. The clustering coefficient of each course node (2 decimal places are reserved).**

| | Sports Physiology | Sports Anatomy | Sports Statistics | Sports Psychology | Sports Introduction | Sports Sociology | Sports Research Methods |
|---|---|---|---|---|---|---|---|
| The clustering coefficient | 0.33 | 0.32 | 0.43 | 0.41 | 0.53 | 0.53 | 0.60 |

of Sports Physiology and other four courses is between 0.4 and 0.61, which is less than the average shortest path length; except for Sports Psychology and Sports Research Methods, the shortest path length between Sports Anatomy and other five courses is between 0.5 and 0.63, which is less than the average shortest path length. The shortest path length between any two courses of the other five courses is mostly longer, and higher than the average shortest path length. Therefore, Sports Physiology and Sports Anatomy are closely related to other courses and play a close role in the mutual support of the courses. Therefore, attention should be paid to the setting and arrangement of the two courses.

According to the betweenness of nodes, it can be concluded from **Table 13** that the node betweenness of Sports Anatomy is the largest, and the two disciplines of Sports Physiology and Sports Anatomy are far more than the other five disciplines. Therefore, Sports Physiology and Sports Anatomy have the intermediary and connectivity between courses in professional learning, and play a connecting role of ability transfer in the learning process.

**Table 14** shows that the clustering coefficient of the whole network is C≈0.39. Among them, the clustering coefficient of Sports Statistics is 0.58, which is larger than the overall network clustering coefficient. Therefore, from 2017 to 2019, Sports Statistics is closely related to the other six courses and has become a supporting course in the course group. Therefore, attention should be paid to the arrangement of teaching hours and contents of the course. The next important courses are Sports Sociology and Sports Research Methods.

Through the grouping analysis of 2009–2016 and 2017–2019, the following changes are found: First of all, when analyzing the course link network map of 2009–2016 and 2017–2019,

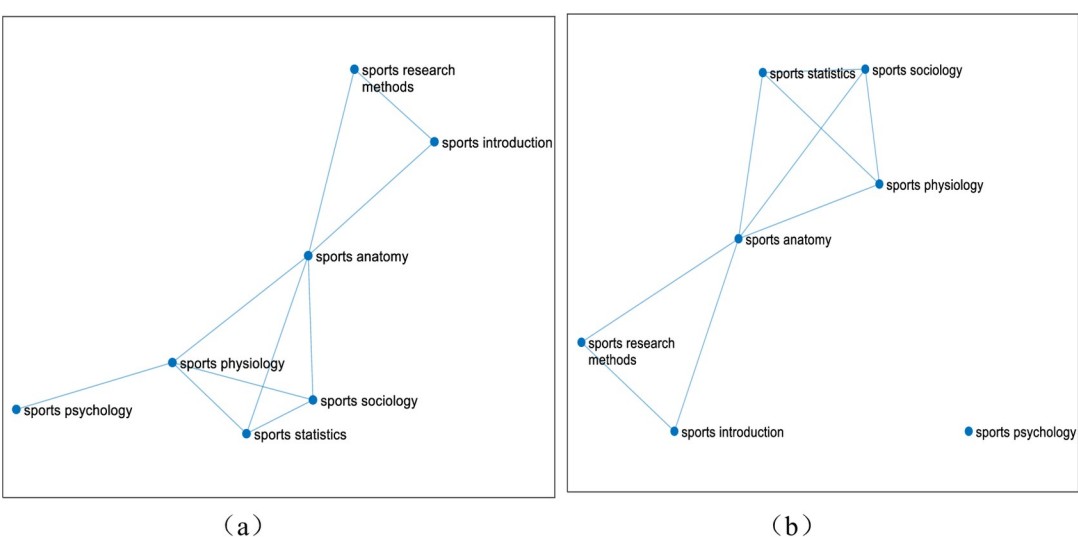

（a）　　　　　　　　　　　（b）

**Fig 6. The linked network of 7 courses of sports major from 2017 to 2019.** (Fig 6 shows that after calculating the correlation of seven professional basic courses of 39 students majoring in sports training and physical education from 2017 to 2019 in East China Normal University, the different course linked network diagrams are got through dynamically changing the threshold K. The seven courses are respectively Sports Physiology, Sports Anatomy, Sports Statistics, Sports Psychology, Sports Introduction, Sports Sociology, Sports Research Methods. The threshold value of figure (a) is 0.4, and the threshold value of figure (b) is 0.5.)

**Table 11. The node degree statistics.**

| Courses | Sports Physiology | Sports Anatomy | Sports Statistics | Sports Psychology | Sports Introduction | Sports Sociology | Sports Research Methods |
|---|---|---|---|---|---|---|---|
| Node degree | 3 | 5 | 3 | 0 | 2 | 3 | 2 |
| Node degree | 4 | 5 | 3 | 1 | 2 | 3 | 2 |

the degree of Sports Anatomy course is the most, which is an important node in the network. It also shows the importance of the course in the course network from 2009 to 2019. In the two stages, the node degree of Sports Physiology and Sports Anatomy is the largest, so from 2009 to 2019, compared with other courses, Sports Physiology and Sports Anatomy are necessary basic courses for professional learning, so the core position has not changed.

Secondly, in the first stage of analyzing the shortest path length and node betweenness, the shortest path length between Sports Anatomy and Sports Statistics and other courses is mostly less than the average shortest path length, so they are closely related and highly intermediary. In the second stage, most of the shortest path lengths between Sports Physiology and Sports Anatomy and other courses are less than the average shortest path lengths, so they are closely related, and have the intermediary and connectivity between courses. The two stages of curriculum change, from Sports Statistics to Sports Physiology. In the continuous development process from 2009 to 2019, compared with Sports Statistics and other courses, Sports Physiology has gradually become more and more closely related, which also shows the specialty of Sports Physiology.

Finally, in the first stage, when calculating the clustering coefficient of network nodes, it is found that among seven professional courses, Sports Research Methods, Sports Introduction, Sports Sociology are more closely compared with other courses. In the second stage, Sports Statistics, Sports Sociology and Sports Research Methods are more closely related to other courses. Compared with the two stages, Sports Research Methods and Sports Sociology are more closely. There is no change in science, but there is a course from the first stage of sports introduction to the course of sports statistics. Sports Research Methods can make students have a general understanding of sports science research. Sports Sociology is a comprehensive subject between sports science and sociology, which is a branch of sociology. It is also a basic discipline in the sports discipline, which is conducive to students' learning of other professional knowledge. Compared with the course of introduction to sports, the Sports Statistics course can affect students' professional level more, and it is an important course for students majoring in sports. Therefore, in the development of the two stages, the role of Sports Statistics course is more obvious, which is also in line with the actual physical education curriculum development.

*4) The analysis combined with the training program.* Based on the course time, credit arrangement and course classification of the seven courses in the undergraduate training

**Table 12. The shortest path length for any two courses (2 decimal places are reserved).**

| | Sports Physiology | Sports Anatomy | Sports Statistics | Sports Psychology | Sports Introduction | Sports Sociology | Sports Research Methods |
|---|---|---|---|---|---|---|---|
| Sports Physiology | 0 | 0.54 | 0.55 | 0.49 | 1.05 | 0.61 | 1.04 |
| Sports Anatomy | 0.54 | 0 | 0.63 | 1.03 | 0.52 | 0.60 | 0.50 |
| Sports Statistics | 0.55 | 0.63 | 0 | 1.04 | 1.14 | 0.74 | 1.13 |
| Sports Psychology | 0.49 | 1.03 | 1.04 | 0 | 1.55 | 1.10 | 1.53 |
| Sports Introduction | 1.05 | 0.52 | 1.14 | 1.55 | 0 | 1.12 | 0.51 |
| Sports Sociology | 0.61 | 0.60 | 0.74 | 1.10 | 1.12 | 0 | 1.11 |
| Sports Research Methods | 1.04 | 0.50 | 1.13 | 1.53 | 0.51 | 1.11 | 0 |

**Table 13. The betweenness of each course node.**

|  | Sports Physiology | Sports Anatomy | Sports Statistics | Sports Psychology | Sports Introduction | Sports Sociology | Sports Research Methods |
|---|---|---|---|---|---|---|---|
| The node betweenness | 3 | 8 | 0 | 0 | 0 | 0 | 0 |

program of this major in East China Normal University, the modeling results are analyzed as shown in **Table 15** below:

Among these seven courses, as a teacher education course, Sports Statistics is designed to make students become excellent PE teachers who can engage in sports scientific research and school sports management, as can be seen from the training scheme of East China Normal University. Sports Introduction, Sports Anatomy, Sports Physiology, Sports Psychology, Sports Research Methods and Sports Sociology are the basic courses of the discipline. They are not only a kind of courses set up by colleges and universities to lay a necessary foundation for the study of specialized courses, but also important compulsory courses for students to master professional knowledge and skills. These seven courses are discussed in detail in this section of the modeling results analysis.

Among the sample size of 462 in the above analysis, modeling network connection diagram and complex network index illustrate that the nodes of "Sports Anatomy" and "Sports Physiology" in the curriculum network are the core nodes of the network, and the credits of these two courses are set as 3 credits, which is the highest of seven courses. It is enough to prove the importance that the training scheme attaches to them. The clustering coefficients of "Sports Anatomy" and "Sports Physiology" in the network nodes are about 0.30 and 0.41 respectively, which are significantly smaller than the average clustering coefficient C. This is because they are in the core of the network.

Besides, known from the node degree, under the condition of threshold k = 0.4, Sports Anatomy has a connected edge relationship with all other courses, while the non-core courses on their connection are not able to be linked to other courses, which highlights the core level of this course. The correctness of this conclusion is also shown in the year analysis. From the training scheme, it can be seen that the semester of "Sports Anatomy" begins earlier than that of "Sports Physiology". As an advanced course, "Sports Anatomy" is in the core position of the network. Thus, it is suggested that students should pay more attention to this course in the process of learning. It is beneficial to the overall transfer of the whole specialized curriculum learning and masters the core courses of the major, which means that the overall learning network transfer can be completed more easily. In the process of students' learning and teaching, highlighting the importance of core courses is also conducive to the construction of students' overall specialized network. According to the modeling results combined with the training scheme, it can be seen that the core levels of the nodes of "Sports Anatomy", "Sports Psychology" and "Sports Physiology" are higher, and the semesters of them begin earlier. According to the above characteristics and the nature of the necessary foundation for the specialized learning, these three courses are the core courses of the small-scale specialized curriculum network, and should be paid enough attention to in the teaching and learning of this major.

**Table 14. The clustering coefficient of each course node (2 decimal places are reserved).**

|  | Sports Physiology | Sports Anatomy | Sports Statistics | Sports Psychology | Sports Introduction | Sports Sociology | Sports Research Methods |
|---|---|---|---|---|---|---|---|
| The clustering coefficient | 0.33 | 0.24 | 0.58 | 0 | 0.50 | 0.57 | 0.52 |

**Table 15. The course schedule and credit.**

| Courses | Sports Physiology | Sports Anatomy | Sports Statistics | Sports Psychology | Sports Introduction | Sports Sociology | Sports Research Methods |
|---|---|---|---|---|---|---|---|
| Semester Course | 2 | 1 | 3 | 4 | 1 | 3 | 6 |
| Credit | 3 | 3 | 2 | 2 | 2 | 2 | 2 |
| Class of course | Basic course | Basic course | Teacher education course | Basic course | Basic course | Basic course | Basic course |

Tip: The course schedule and credit data are inquired from training programs of East China Normal University.

The connected edge relationship of the network is constructed based on curriculum relevance. And the calculation results of curriculum relevance show that the two courses," Sports Physiology" and "Sports Anatomy", have the highest correlation. "Sports Anatomy", which is based on human anatomy to studies the influence of sports on morphology, structure, growth and development of human body, is a branch of human anatomy. It is an important basic course and advanced course in sports human science. "Sports Physiology" studies the structural and functional changes of human body under the influence of sports activities and sports training. The beginning time of "Sports Anatomy" is the first semester, while that of "Sports Physiology" is the second semester. The transfer process from "Sports Anatomy" to "Sports Physiology" is a learning process from figurative morphological structure to abstract functional structure of human system. These two courses are so closely related, which means that in the whole seven courses, the learning transfer between Sports Anatomy and Sports Physiology is the easiest to complete, and mastering the "Sports Anatomy" as an advanced course can make it more convenient to use knowledge transfer to learn "Sports Physiology". In this part of the teaching process, attention should be paid to the foreshadowing role of the advanced course, while avoiding the negative transfer effect of the advanced course in the learning process of the later courses.

To sum up, among the specialized courses composed of these seven courses, "Sports Anatomy" is not only the core foundation of the course network, but also the advanced courses of specialized learning (the opening time is the first semester). It means that in the process of teaching, the importance of this course should be highlighted, so as to achieve the reasonable use of learning transfer to complete the entire specialized learning.

## 4.2 Permeation mechanism of learning transfer in multiplex complex networks

Through the analysis of the topological structure of the complex network of learning transfer, this paper investigates the permeation law of learning transfer among different courses of each major on the basis of deeply interpreting the learning transfer mechanism between the courses related to the same major in the network. In this part, mainly based on the complex network of courses in each major, the multiplex networks of learning transfer are constructed by coupling the network of courses in different majors through the same course. And the permeation effect of learning transfer among different majors is characterized by the linear diffusion law formula (1) of multiplex complex networks. The link network of eight courses for two physical education majors is shown in **Fig 7**.

Couple the two networks of the two majors by establishing the inter-layer connection within the same 7 courses of the two majors. Node No. 1 is a specialized course of sports training, and the network sample size is 218. Node No.2 is the physical education specialized

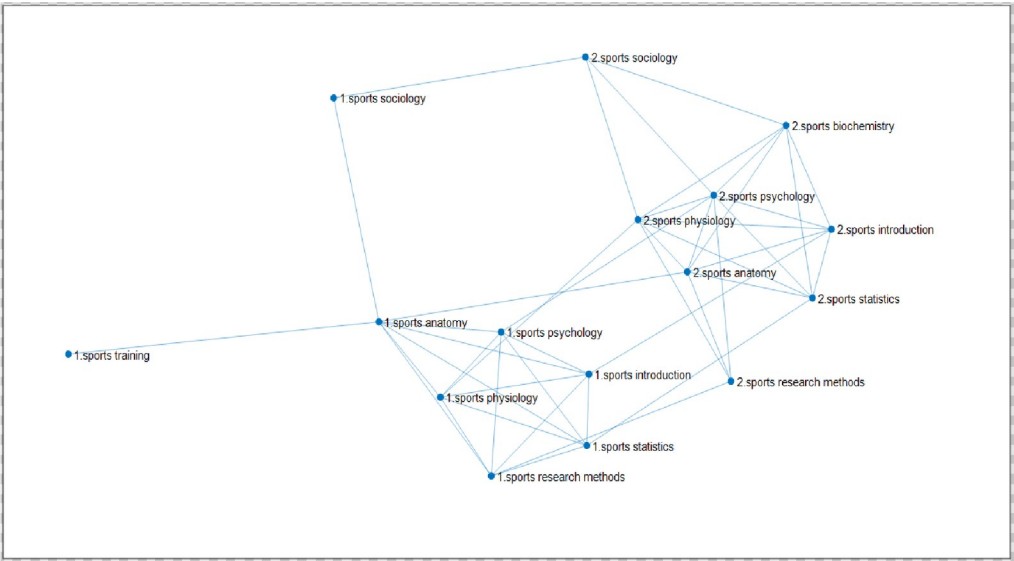

**Fig 7. Link network of eight courses for two physical education majors.** (Fig 7 is a multiple network of learning transfer for sports training and physical education majors of East China Normal University through the same curriculum coupled with different professional curriculum networks. Among them, there are 7 courses of the same two majors, which are Sports Physiology, Sports Anatomy, Sports Statistics, Sports Psychology, Sports Introduction, Sports Sociology, Sports Research Methods, and different courses are "Sports Training Major—Sports Training" and "Physical Education Major—Sports Biochemistry". 218 samples of sports training and 244 samples of physical education were selected.").

course, and the network sample size is 244. In **Fig 7**, the establishment of connected edge relationship within the major is based on the similarity of courses. That is, the construction principle of the single-layer specialized network is the same as that of 4.1.1 single-layer network; the establishment of connected edge relationship between different majors is based on the same course score.

The interlayer diffusion coefficient $D_{\alpha,\beta}$ is constructed by summing the ratio of the average score of the courses of the sports training major (label 1) and the average score of the same courses of the physical education major (label 2) and dividing by the same number of courses, that is, the formula (2) of interlayer diffusion coefficient of linear diffusion from sports training major to physical education major.

$$D_{12} = \left( \sum_{i=1}^{N} \frac{mean(x_1^i)}{mean(x_3^i)} \right) \times \frac{1}{N} \tag{2}$$

Where $mean(x_1^i)$ represents the average score of the course of the first layer, which is the network layer of sports training major (label 1), meanwhile $mean(x_3^i)$ represents the average score of the courses of the second layer, which is the network layer of sports education major (label 2), and N means the number of same courses. Besides, the principle of learning transfer permeation effect between two majors is characterized by the linear diffusion algorithm of multiplex networks by calculating the second smallest eigenvalue reaction diffusion effect of coupled multilayer network.

The higher the second small eigenvalue is, the easier it is to transfer learning among different majors.

From the perspective of professional teaching, this is because there is no obvious difference between the two majors in the learning requirements of Sports Research Methods. There is no

distinguishing feature between the two majors. Therefore, both the students majoring in sports training and those in physical education can achieve the same learning objectives when learning the course. Therefore, it is easy to complete the course of Sports Research Methods from sports training to physical education. As shown in **Table 16**, the results above show that in the transfer process from the sports training major (label 1) to the physical education major (label 2), "Sports Research Methods" is the easiest to diffuse, that is, the easiest to complete the transfer. From the perspective of professional teaching, this is because there is no obvious difference between the two majors in the learning requirements of Sports Research Methods. There is no distinguishing feature between the two majors. Therefore, both the students majoring in sports training and those in physical education can achieve the same learning objectives when learning the course. Therefore, it is easy to complete the course of Sports Research Method from sports training to physical education.

The sixth course "Sports Sociology" will be the one that hinders most the transfer from sports training to physical education. According to the training goal of the training scheme for students, the sports training major is to cultivate applied talents who can be competent in sports schools, primary and secondary schools and fitness clubs. While, the sports education major aims to train excellent physical education teachers who can be qualified in physical education teaching, extracurricular physical education training and competition in primary and secondary schools, and can engage in physical education scientific research and school physical education management. Sports Sociology is a branch of sociology, which is closely related to teacher education. The difference of training objectives leads to the poor learning of Sports Sociology in the teaching of sports training major and in the learning process of students. However, the course in physical education pedagogy occupies an important position. It means that it is necessary to focus on the supplementary knowledge of the course to complete the transfer from sports training major to sports education major. The results of the transfer from sports training major to physical education major show that the current learning situation of 2 of the same 7 courses will hinder the diffusion of the overall network, but the hindrance is not great, that is, it is still easy to complete the overall transfer from sports training major to physical education major, and attention should be paid to the cultivation of humanistic quality and scientific literacy.

In addition, in order to clarify and study the robustness of the current findings, with overall promotion as the starting point and problem-solving as the focus, we pay attention to quality, and actively promote the standardization of the research. Based on the original experiment, the group study is further carried out. Taking the average score as the dividing line, it is divided into two groups: one with above average scores and one with below average scores. The average scores of students in 7 courses of two majors (218 samples of sports training and 244 samples of Physical Education) are calculated respectively, and then the sample size of students whose scores are higher than or lower than their average scores are sorted out, and the scores higher than average scores and lower than average scores are taken as intersection, and the results of 7 courses of two majors are higher than average scores and the average score of the sample size, and then the sample size of the two majors higher than the average score to build a complex network diagram, through the same course coupled with different

**Table 16. The second smallest eigenvalue of the transfer network from major 1 to 2.**

|  | Sports Physiology | Sports Anatomy | Sports Statistics | Sports Psychology | Sports Introduction | Sports Sociology | Sports Research Methods | Sports Biochemistry |
|---|---|---|---|---|---|---|---|---|
| The second smallest eigenvalue | $1.70e^{-1}$ | $6.59e^{-3}$ | $1.78e^{-1}$ | $1.70e^{-1}$ | $1.78e^{-1}$ | $-8.42e^{-1}$ | $1.86e^{-1}$ | $-4.69e^{-2}$ |

 

professional curriculum network to build a multi-network of learning transfer, the sample size of two majors with lower than average score is the same. For sports training majors, the sample size higher than the average score is 94, and the sample size lower than the average score is 91; The sample size of the physical education major with a higher than average score is 116 and the sample size with a lower than average score was 110. The course network diagram constructed is as follows:

**Fig 8** shows that the link network of 8 courses of two physical education majors is higher than the average score, and the establishment of the connection relationship is the same as that in Fig 7. The diffusion coefficient between the average score of sports training (label 1) and that of physical education (label 2) is divided by the same number of courses, and then the diffusion coefficient can be calculated according to the linear diffusion formula of multiple networks. The second characteristic value of the network is transferred from sports training major (label 1) to physical education major (label 2), as shown in **Table 17**.

Table 17 shows that, with higher average scores, in the process of transferring from sports training major (label 1) to physical education major (label 2), Sports Sociology is the easiest course to complete the transfer. On the one hand, the students with higher than average scores have stronger learning ability, while compared with other disciplines, Sports Sociology has lower learning difficulty. On the other hand, the two majors are more interested in this course. The learning requirements are the same, and the learning objectives of students are the same. Therefore, in the process of transferring from sports training major (label 1) to physical education major (label 2), Sports Sociology is the easiest to complete the transfer. Among the 8 courses, Sports Introduction, Sports Statistics, Sports Psychology, Sports Scientific Research Methods and Sports Biochemistry are the courses that hinder the transfer of sports training to physical education when their scores are higher than average. However, in the case of higher than average scores, the blocking effect is not great. It is easy to transfer the whole specialty.

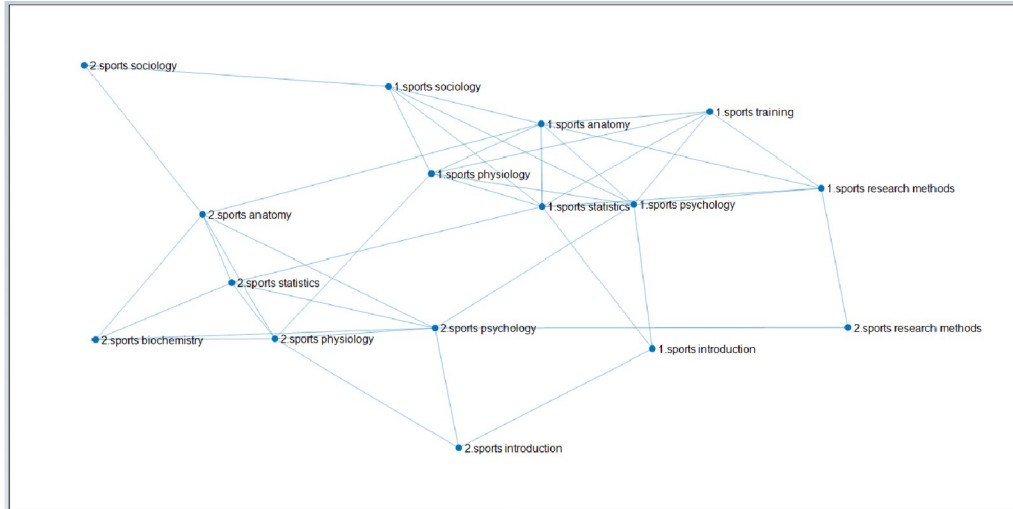

**Fig 8. The linked network of 8 courses with higher than average scores of students majoring in physical education.**
(Fig 8 is a multiple network of learning transfer for sports training and physical education majors in East China Normal University through the same curriculum coupled with different professional curriculum networks. Among them, there are 7 courses of the same two majors, which are "Sports Physiology, Sports Anatomy, Sports Statistics, Sports Psychology, Sports Introduction, Sports Sociology, Sports Research Methods", and the different courses are "Sports Training Major-Sports Training Science" and "Sports Education Major-Sports Biochemistry". The selected data are the sample size of students whose scores are higher than average, 94 samples of sports training major and 116 samples of physical education.).

**Table 17. The second smallest eigenvalue of the transfer network from major 1 to 2.**

| | Sports Physiology | Sports Anatomy | Sports Statistics | Sports Psychology | Sports Introduction | Sports Sociology | Sports Research Methods | Sports Biochemistry |
|---|---|---|---|---|---|---|---|---|
| The second smallest eigenvalue | $4.42e^{-2}$ | $3.91e^{-2}$ | $-5.45e^{-4}$ | $-1.45e^{-2}$ | $-1.24e^{-1}$ | $2.19e^{-1}$ | $-1.45e^{-1}$ | $-1.88e^{-2}$ |

The above is the specific analysis of the above-average score. Compared with the situation of higher average score, the details of lower than average score are shown in **Fig 9**.

Fig 9 shows that the link network of 8 courses of two sports majors is below the average score. Fig 8 shows that the second small characteristic value of the network can be transferred from sports training major (label 1) to physical education major (label 2), as shown in **Table 18** below:

Table 18 shows that, with below-average scores, in the process of transferring from sports training major (label 1) to physical education major (label 2), because Sports Biochemistry is only a course of physical education major, so after excluding the course, it can be seen from the above analysis that Sports Sociology is a core course in learning transfer and the learning difficulty is relatively low. Therefore, the two majors have basic requirements for this course Therefore, Sports Sociology is the easiest to complete the transfer of a course, and the rest of the courses, Sports Introduction, Sports Statistics, Sports Psychology, Sports Research Methods, Sports Anatomy, Sports Physiology are courses that hinder the transfer of sports training to physical education under the condition of lower than average score. However, because the second eigenvalues are small, the hindrance is not great, and the transfer of learning between majors is easier. From the above two cases of above-average scores and below-average scores, the overall transfer from sports training major to physical education major is relatively easy.

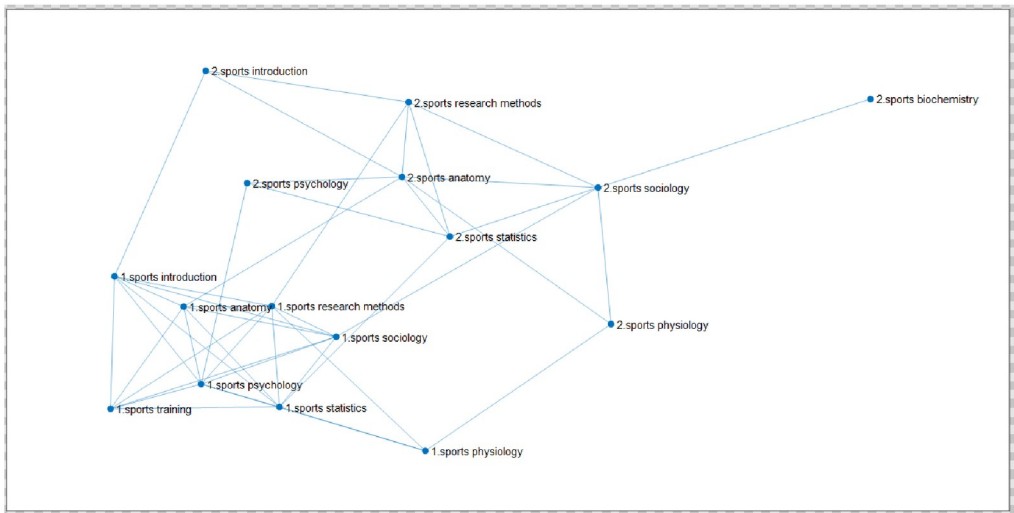

**Fig 9. The linked network of 8 courses with lower than average scores of students majoring in physical education.** (Fig 9 shows the multiple network of learning transfer for sports training and physical education majors in East China Normal University through the same curriculum coupled with different professional curriculum networks. Among them, there are 7 courses of the same two majors, which are "Sports Physiology, Sports Anatomy, Sports Statistics, Sports Psychology, Sports Introduction, Sports Sociology, Sports Research Methods", and the different courses are "Sports Training Major Sports-Training Science" and "Sports Education Major-Sports Biochemistry". The selected data are the sample size of students whose scores are lower than average, 91 samples of sports training major and 110 samples of physical education.).

**Table 18. The second smallest eigenvalue of the transfer network from major 1 to 2.**

|  | Sports Physiology | Sports Anatomy | Sports Statistics | Sports Psychology | Sports Introduction | Sports Sociology | Sports Research Methods | Sports Biochemistry |
|---|---|---|---|---|---|---|---|---|
| The second smallest eigenvalue | $-6.38e^{-2}$ | $-1.10e^{-2}$ | $-1.74e^{-2}$ | $-3.13e^{-2}$ | $-2.53e^{-2}$ | $2.45e^{-2}$ | $-1.72e^{-2}$ | $1.42e^{-1}$ |

After the observation of Figs 8 and 9, to enhance the scientific and rigorous nature of the research, for the sports training major and physical education major, the complex network diagram of the same course in different majors can be constructed by further analyzing the difference between the two professional classes. Among them, the sample size of male students in sports training major is 120, while that of female students is 98. The sample size of male students in physical education majors is 199 and that of female students is 45. Figs 10 and 11 can be obtained for the link network diagram of 8 courses in two different majors. Based on the formula of the interlayer diffusion coefficient, the second small eigenvalue of the multi-layer network between the two different majors is obtained, and the diffusion effect is analyzed according to the second small eigenvalue.

By comparing **Figs 10** and **11**, the differences between male and female students' learning transfer can be more intuitively seen, which is conducive to the development of personalized teaching strategies for students of different genders, so as to build a more perfect teaching system. The course network diagram constructed is as follows:

Fig 10 is the link network of 8 courses of two sports majors made according to male students' grades. The principle is the same as That in Fig 9. Furthermore, the interlayer diffusion coefficient can be constructed, and then the second small eigenvalue of the transfer network from sports training major (label 1) to physical education major (label 2) can be calculated according to the linear diffusion formula of multiple networks, as shown in **Table 19** below.

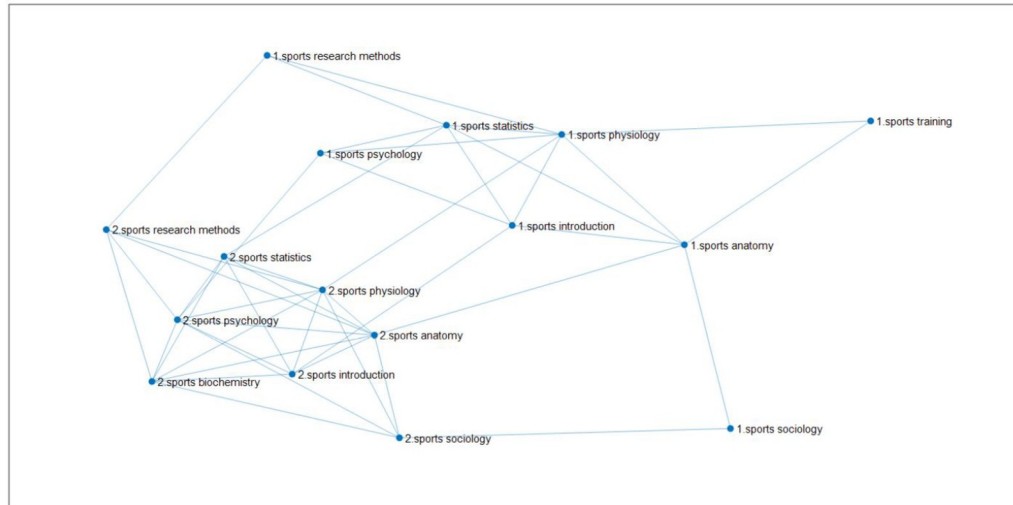

**Fig 10. The linked network of the scores of 8 courses for male students majoring in physical education.** (Fig 10 shows the multiple network of learning transfer for sports training and physical education majors in East China Normal University through the same curriculum coupled with different professional curriculum networks. Among them, there are 7 courses of the same two majors, which are "Sports Physiology, Sports Anatomy, Sports Statistics, Sports Psychology, Sports Introduction, Sports Sociology, Sports Research Methods", and the different courses are "Sports Training Major Sports Training Science" and "Sports Education Major Sports Biochemistry". The selected data are the sample size of boys' performance, 120 samples of sports training major and 199 samples of physical education major.).

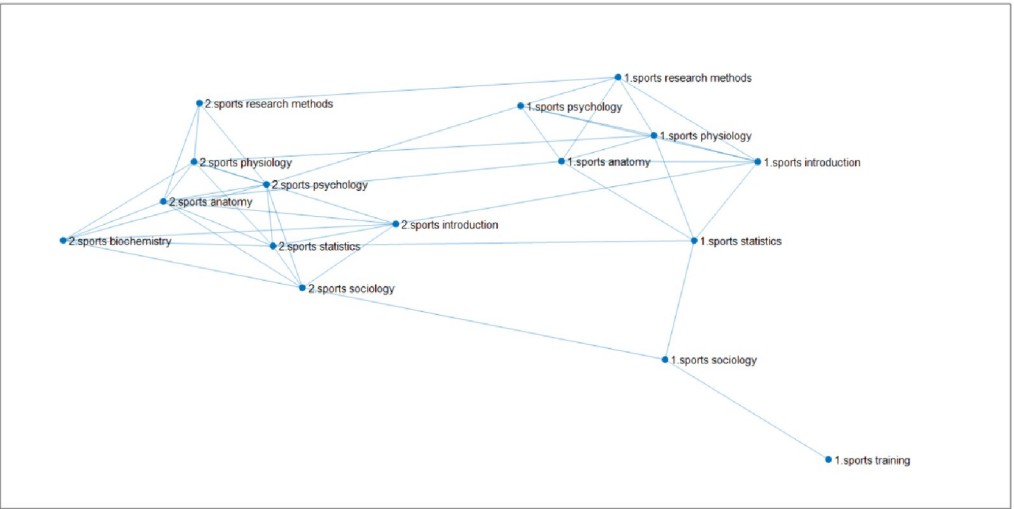

**Fig 11. The linked network of the scores of 8 courses for female students majoring in physical education.** (Fig 11 is a multiple network of learning transfer for sports training and physical education majors in East China Normal University through the same curriculum coupled with different professional curriculum networks. Among them, there are 7 courses of the same two majors, which are "Sports Physiology, Sports Anatomy, Sports Statistics, Sports Psychology, Sports Introduction, Sports Sociology, Sports Research Methods", and the different courses are "Sports Training Major Sports Training Science" and "Sports Education Major Sports Biochemistry". The selected data are the sample size of female students, 98 samples of sports training major and 45 samples of physical education.).

It can be seen from Table 19 that in the process of transfer from sports training major to physical education major, Sports Sociology is the easiest to complete the transfer. For the remaining seven subjects, although the second smallest eigenvalue is not very large, the blocking effect is not significant.

Fig 11 is the link network of 8 courses of two sports majors made according to girls' grades, and the principle is the same as Fig 9. At the same time, the interlayer diffusion coefficient is constructed, and then according to the linear diffusion formula of multiple networks, the second small eigenvalue of the transfer network from sports training major (label 1) to physical education major (label 2) can be calculated, as shown in **Table 20** below.

In the second small eigenvalue table (Table 20) of the transfer network from Major 1 to Major 2 based on the sample of female students, it can be concluded that the second small eigenvalue of the transfer network of Sports Biochemistry is the largest, so this discipline is the easiest to complete the transfer learning from sports training major to physical education major. The second smallest characteristic value of Sports Sociology is the second, which is easy to complete the transfer learning from sports training major to physical education major. And the table shows that the remaining five disciplines have little hindrance.

## 5. Conclusion and prospect

Under the guidance of the complex network theory, based on the academic achievement data of college students, this project constructs a single-layer complex network and multiplex

**Table 19. The second smallest eigenvalue of the transfer network from major 1 to 2.**

|  | Sports Physiology | Sports Anatomy | Sports Statistics | Sports Psychology | Sports Introduction | Sports Sociology | Sports Research Methods | Sports Biochemistry |
|---|---|---|---|---|---|---|---|---|
| The second smallest eigenvalue | $-1.38e^{-1}$ | $9.16e^{-2}$ | $-1.63e^{-1}$ | $-1.95e^{-1}$ | $-1.35e^{-1}$ | $8.62e^{-1}$ | $-2.84e^{-1}$ | $-3.78e^{-2}$ |

**Table 20. The second smallest eigenvalue of the transfer network from major 1 to 2.**

|  | Sports Physiology | Sports Anatomy | Sports Statistics | Sports Psychology | Sports Introduction | Sports Sociology | Sports Research Methods | Sports Biochemistry |
|---|---|---|---|---|---|---|---|---|
| The second smallest eigenvalue | $-2.08e^{-1}$ | $-2.05e^{-1}$ | $-4.50e^{-2}$ | $-2.40e^{-1}$ | $-2.02e^{-1}$ | $4.36e^{-1}$ | $-2.57e^{-1}$ | $7.21e^{-1}$ |

complex networks for learning transfer. Besides, through the research on the topological structure and evolution law of the network, the essential law of learning transfer is clearly defined and recommendations are made for the optimization of teaching strategies. Compared with the traditional education data analysis based on statistics, the complex network analysis method, by introducing the perspective of complex network research into the field of learning transfer, can be better adapted to the era of big data in education, which opens up new ideas for the research in this field. The complex network of learning transfer provides a new perspective for discovering the overall structure and hidden connections of the course in the process of learning transfer. The main contributions of this study are as follows:

1. The rationality of using the complex network to study learning transfer is verified, and a complex network suitable for measuring learning or skills transfer in the process of physical education teaching is constructed.

2. The complex network of learning transfer provides a new perspective for discovering the overall structure and hidden connections of the course during learning transfer. Based on this network, the core curriculum can be quantified, and the continuity between courses, clustering characteristics and the basic mode of learning transfer can be found. By analyzing the total data set and grouping analysis of the two stages from 2009 to 2016 and from 2017 to 2019, the significance of learning transfer becomes more obvious. This also provides a platform for exploring the differences in the curriculum structure of different majors and the learning transfer between male and female students.

3. The relevance of intra-layer and inter-layer in learning transfer network is reasonably measured, and the permeation and diffusion mechanism of the curriculum network of learning transfer constructed by different related indexes are excavated.

4. It is found that the mastery of the core courses in the same specialized curriculum network will positively promote the learning of most other courses in the learning network, and the influencing factors of cross-major transfer mainly come from the different degrees of students' mastering courses required by different majors.

In this paper, linear correlation coefficients, such as rank correlation coefficient, are mainly used to evaluate curriculum correlativity. In order to evaluate curriculum correlation more objectively and comprehensively, cosine distance, mutual information and other more correlation indexes will be used to depict the nonlinear correlation and dependency correlation of curriculum relevance in the future. More detailed and targeted collection is needed in order to reflect the actual situation more comprehensively and objectively quantity data, so that the application of the model can be further optimized. If conditions permit, social research such as expert interviews and document surveys, combined with survey results and the existing quantitative analysis, would put forward more targeted suggestions. Subsequent researches can also combine the time sequence of the course to construct temporal networks to explore the time sequence evolution rule of learning transfer permeation mechanism. In addition, from the perspective of students, a network of college students can be built by using students' academic achievements. Through the theory of community division of complex network, the paper

analyzes the differences of college students' abilities, provides a quantitative analysis basis for discovering the hidden patterns among students, and offers a reference for education management departments to carry out personalized teaching.

## Supporting information

**S1 File. Data availability statement.**
(DOCX)

## Author Contributions

**Conceptualization:** Xin Feng, Shuhui Hu.

**Data curation:** Long Chen, Nan Wang.

**Formal analysis:** Jiapei Li.

**Funding acquisition:** Xin Feng.

**Investigation:** Xin Feng, Yi Zhao.

**Methodology:** Jiapei Li.

**Software:** Shuhui Hu, Nan Wang.

**Validation:** Yi Zhao, Long Chen.

**Visualization:** Shuhui Hu, Nan Wang.

**Writing – original draft:** Xin Feng.

**Writing – review & editing:** Yi Zhao.

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
