## [Decision Letter · Decision Letter 0]

10 Aug 2020

PONE-D-20-07521

An approach to the Permeation Mechanism of Learning Transfer and Teaching Strategy in Physical Education Based on Complex Network

PLOS ONE

Dear Dr. Zhao,

Thank you for submitting your manuscript to PLOS ONE. After careful consideration, we feel that it has merit but does not fully meet PLOS ONE’s publication criteria as it currently stands. Therefore, we invite you to submit a revised version of the manuscript that addresses the points raised during the review process.

We look forward to receiving your revised manuscript.

Kind regards,

Carlos Gracia-Lázaro

Academic Editor

PLOS ONE

Additional Editor Comments:

Please, take into account the comments, concerns, and remarks of both reviewers. In particular, reviewer 1 exposes serious concerns that have to be properly addressed.

Journal Requirements:

Reviewers' comments:

Reviewer's Responses to Questions

**Comments to the Author**

1. Is the manuscript technically sound, and do the data support the conclusions?

Reviewer #1: No

Reviewer #2: Partly

2. Has the statistical analysis been performed appropriately and rigorously? 

Reviewer #1: No

Reviewer #2: Yes

3. Have the authors made all data underlying the findings in their manuscript fully available?

Reviewer #1: No

Reviewer #2: Yes

4. Is the manuscript presented in an intelligible fashion and written in standard English?

Reviewer #1: No

Reviewer #2: Yes

5. Review Comments to the Author

Reviewer #1: In their manuscript titled “An approach to the Permeation Mechanism of Learning …”, the authors Xin et al, study diffusion of learning across courses in two majors, namely, sports education and sports training. In their analysis, they construct a network, primarily taking into correlation of scores and then ratio of scores between majors providing a multiplex description.

The findings in the manuscript could be relevant to the research communities in learning theory or organisational behaviour. Also, the demonstration of the general idea of diffusion on networks constructed at the level of courses (“knowledge unit”), and not individuals, seems interesting. However, with the current manuscript, several criticisms are in order.

First, the overall presentation of the manuscript is rather sloppy. The introduction to the problem, the framing, the methods and analyses, all lack clarity. More so, because of the errors that are present in places like the equations 1 and 2, and elsewhere. The equation 1 and the formula for D_12 (written D_13), both need to be carefully explained. Equation 1 is well known but is not supported by a reference (like ref.17). In fact, the authors did not put a citation beyond page 4. Also, not sure why the authors use the word “infiltration”.

What are the motivations behind choosing the given (eq. 2) form for the inter-layer diffusion coefficient?

Is there a significance of clustering, or the triangulations, in general? The existence of a strong link between Sports Anatomy and Sports Physiology is hardly surprising and as the authors themselves say, “...it can only provide microscopic and partial course information and transfer behavior”.

The authors should either provide reasons for choosing the correlation threshold as 0.4, for instance - a threshold where one would get a spanning tree; or demonstrate the variation/sensitivity of the other quantities on this threshold.

I suggest that the authors look at the following additional analyses to elucidate and study the robustness of the current findings, given the fact that the sample-set is heterogeneous: (a) Divide the dataset into two (or more) non-overlapping year-spans (within 2009-14) and construct the corresponding networks; (b) Divide each of the 202 and 198 samples into two (or more) percentile ranges and perform the analysis. That is, compare the networks while considering students with either above or below average scores. In the latter case, comparing the gender aspects (male/female students) might be useful.

In its current form I do not recommend publication of manuscript.

Reviewer #2: This paper analyses an application of network science in teaching strategies.

The idea sounds interesting, however some issues must be addressed

before this paper can be accepted for publication:

1) The abstract seem a bit confusing, with repetition of terms.

Please improve the readability of it. Results are not even mentioned

or highlighted.

2) Related works section should be improved.

There are some applications including cognitive

networks, knowledge acquisition and education. There are

some related works that have not been mention (see e.g.

doi: 10.1209/0295-5075/113/28007 doi: 10.1016/J.INS.2017.08.091

and doi: 10.1038/s41598-018-20730-5).

3) Figure quality should be improved. Most of them are very hard to read

given the low resolution.

4) Text should be improved. There are many errors that are definetily not

acceptable. For example: "Figure3". There are no space between punctuation marks

and words, and so on.

5) Why decimal places are important? In the main manuscript the authors provided

5 decimal places. Is this really necessary?

6) Results should be better discussed. What is the implication of a node having a high degree? How about assortativity.

6. PLOS authors have the option to publish the peer review history of their article (what does this mean?). If published, this will include your full peer review and any attached files.

Reviewer #1: No

Reviewer #2: No

---

## [Author Response · Author response to Decision Letter 0]

13 Nov 2020

Dear editors and reviewers:

 Thank you very much for giving us an opportunity to revise our manuscript. We appreciate the editor and reviewers very much for their constructive comments and suggestions on our manuscript entitled “An approach to the Permeation Mechanism of Learning Transfer and Teaching Strategy in Physical Education Based on Complex Network”. We have studied reviewers’ comments carefully. According to the reviewers’ detailed suggestions, we make the following explanations on the explanation and modification of the relevant contents of our paper.

Reviewer #1:

1.First, the overall presentation of the manuscript is rather sloppy. The introduction to the problem, the framing, the methods and analyses, all lack clarity. 

Problem.

First of all, traditional research is mostly based on the basic hypothesis of learning transfer theory, and uses education statistics-related methods to conduct correlation and regression analysis on various problems in the transfer process, which can certainly provide some meaningful references, but it can only provide microscopic and local course information and transfer behavior, and cannot comprehensively consider learning and transfer from a global, dynamic and complex perspective. In addition, the overall course structure, course relationships, and student professional associations are often not considered, and the patterns and abilities of student transfer cannot be analyzed. Thus, the following specific research questions are posed in this paper.

(1) How to bring the perspective of complex network research into the field of learning transfer and deeply analyze and dig into the data of college students' academic performance, and what are the application prospects and teaching improvement significance of learning transfer? 

(2) What is the overall structure of the curriculum and the hidden connections in the learning transfer process, when considered systematically and quantitatively from a global, dynamic, network perspective? (3) How can we discover the hidden patterns among students, explore the evolution of college students' learning career, and implement personalized teaching and tailor-made teaching?

Framework.

Our careful examination revealed that the article frame was indeed unclear, and we thank the jury experts for correcting it for us. We have also sorted out the framework of the article. The logical summary of the specific framework of the paper is as follows.

(1) The introductory section briefly describes the background to the study of learning transfer, based on a multilayer complex network learning transfer penetration mechanism

(2) Surveyed relevant research reviews and identified the specific research direction of this paper. 

(3) Empirical data processing and basic statistical analysis were carried out.

(4) A model of learning transfer in single-layer complex networks and a model of learning transfer penetration mechanisms in multiple complex networks were developed and analyzed and discussed. 

(5) Draw conclusions and future perspectives from the above research.

Methods.

Complex network is the theory of the study of complex system structural features, dynamic evolution and other laws, to some extent to make up for some of the traditional research shortcomings, with nodes and nodes in a certain time scale of intricate relationships between the network structure to explore its structural features and evolutionary laws, complex network in the learning transfer of the edge is a substantial internal link in the objective embodiment, is quantifiable, can be Computational. Learning transfer can be modeled and studied at the system level from a dynamic and quantitative perspective, reflecting both the scientific and superiority of complex networks for learning transfer research and, to some extent, the nature of learning transfer networks. The specific research methods are summarized as follows.

(1) To investigate the learning transfer that exists in different courses in different situations, a single-level complex network for the same course and a multi-level complex network for different majors are constructed separately.

(2) In the single-layer complex network of courses, the correlation of course variables is depicted by the correlation of grades using real students' grades as the sample, and finally the correlation of courses determines the connected relationship of the complex network. Single-level complex networks use qualitative analysis as well as quantitative complex network indicators to investigate the mechanisms of learning transfer propagation within the network.

(3) Based on the successful construction of a single-layer network, different single-layer networks are coupled together to form a multi-layer complex network, and the linear diffusion formula of the multi-layer complex network is used to investigate the inter-network penetration propagation mechanism of learning transfer.

(4) In this paper, the percolation mechanism of skill transfer in learning transfer is explored using the real coursework results of students from the Physical Education Department of East China Normal University as a sample, and the patterns observed in learning transfer are based on this.

Analysis:

(1) Descriptive statistical analysis.

In order to better see the specific distribution of the course grades, the normal distribution test was carried out, taking Introduction to Sports and Sports Research Methods as examples. The overall situation is better than that of physical education majors, and the deficiencies analyzed by the mean are that it does not reflect the individual characteristics of each student, and a systematic course relationship cannot be obtained, and the sample size is small, which may not fully reflect the current situation.

(2) Single-layer network construction.

① The first construction of a single complex network, the correlation between different courses portrayed by the rank correlation coefficient of grades, set K indicates the threshold of correlation, course correlation greater than K, there is an edge relationship between courses, otherwise not connected. By setting different thresholds, dynamically show the rich correlation between different courses. The greater the number of a course edge, the more prominent its nodes in the network is the core degree, that is, the course in the entire network of courses is more important.

②The shortest path length between nodes can visually depict the degree of closeness between courses, reflecting the ability to migrate between courses, the analysis knows that the longer paths are from the core degree of course combination. 

③The data set is divided into 2009-2010, 2011-2013, 2014-2016, 2017-2018, and 2019 multiple non-overlapping years to construct a single-level network to analyze the structural relationships between courses in different program structures and the degree of core in their networks, respectively. 

④ The importance of the core curriculum in exercise anatomy and exercise physiology can be highlighted from the side by the comparison of the clustering coefficients in the fourth part of the latter section. 

⑤ Combined with the analysis of the training program, Sports Anatomy is both the core curriculum of the athletic training program and the core curriculum of the physical education program needs more attention.

(3) Construction of multiple networks. On the basis of the above construction of multiple networks, through the two majors' course averaging sum divided by the same number of courses to get the inter-layer diffusion coefficient using the linear diffusion formula to get the second small eigenvalue to reflect the results of diffusion. The larger the value, the easier it is to diffuse transfer, sports research methods are the easiest to complete, while the sociology of physical education is the most difficult to complete the transfer, because of the different training goals of different professions, physical education focused on training excellent teachers, sociology of physical education belongs to the education courses, more core, and sports training focused on training applied talents, the core of the course is not enough, so the most difficult to complete the transfer. The same seven courses in six courses will hinder the spread of the overall network, all due to this reason, in addition, the two majors were sorted out to obtain seven courses above average and average student performance sample size, and then the above average sample size in the two majors to build a complex network diagram, through the same course coupled with the network of different professional courses to build a multiple network of learning transfer, two majors below the average of the average of the two courses. The sample size of average scores is the same. Observe changes in two complex network diagrams against student gender in the sample size and analyze the association between inter-course network link diagrams and student gender for above or below average scores.

2.More so, because of the errors that are present in places like the equations 1 and 2, and elsewhere. The equation 1 and the formula for D_12 (written D_13), both need to be carefully explained. Equation 1 is well known but is not supported by a reference (like ref.17). In fact, the authors did not put a citation beyond page 4. 

After careful examination, we find that equation (1) and equation (2) are the same formula, but they are marked in two ways. Thank the reviewers for their careful review. The references supporting Formula 1 are the original reference 16 instead of reference 17, which is now reference 19 due to changes in the references.

In order to better understand the meaning of Formula 1, we supplement the places not specified in the formula: in Formula 1, the meaning of M is the number of all network layers in the multiple network, n is the total number of network nodes, the value range of node I is from 1 to N, and the value range of α in layer α is from 1 to m (the meaning of β is the same as α), and (T) represents the state of node i in layer α;

② Due to our carelessness, the formula originally written as D12 was written as D13, and the coefficient calculated here is the coefficient between the first layer of sports training professional network layer and the second layer of physical education professional network layer, so it should be D12. Now the formula has been revised. We sincerely thank you for your carefulness!

3.Also, not sure why the authors use the word “infiltration”.

According to the theory of educational psychology, learning transfer is the influence of one kind of learning experience on another (or the ability to transfer the experience learned in one situation to a new situation). Infiltration often appears in the Chinese context, on the one hand, it means a subtle process. Especially from the perspective of education, the impact on different professional courses can not be seen immediately, it is not instant, but chronic diffusion. On the other hand, infiltration is a two-way process, for example: A has an impact on B, B has an impact on C, and they interact and interweave with each other.

In the process of learning transfer, sports anatomy studies the influence of sports on the human body's morphological structure and its laws, while sports physiology studies the changes of human body's structure and function under the influence of sports activities and sports training, studies the law of functional changes of human body in the process of exercise, and the physiological law of forming and developing sports skills. The two perspectives are different, but in the learning process The knowledge between the two is mutual penetration and integration, and the infiltration of learning transfer behavior helps to improve students' understanding of the curriculum. It is not only a kind of transfer in the process of learning, but also a kind of transfer.

Because we use the word "infiltration" in the Chinese context, it may lead to the loss of some semantics in the process of translation into English, resulting in the inability to fully express the original meaning. In addition, among the valuable suggestions given by experts, we think that interaction can better describe the interaction between layers and different professions, and "infiltration" can be changed into interaction.

4.What are the motivations behind choosing the given (eq. 2) form for the inter-layer diffusion coefficient?

The diffusion coefficient originally represents the physical quantity of gas (or solid) diffusion degree, but in this paper, the interlayer diffusion coefficient represents the diffusion ability between different layers in multi-layer complex network, which can quantitatively and specifically reflect the diffusion ability between different layers. First of all, the main reason why we set the interlayer diffusion coefficient is the influence between different specialties and the influence of different disciplines within the specialty. The influence of these two kinds of influences is different. Even if the same course in different professional teaching process, its emphasis and role is not the same.

For example, in the original text, we calculate the diffusion coefficient and couple the second eigenvalue of the multilayer network to reflect the diffusion effect. Among the eight subjects in the data set, the second smallest eigenvalue of sports training major to physical education major is similar and easy to transfer, but the second smallest eigenvalue of "Sports Sociology" from sports training major to physical education major is the lowest, which is -0.8421. This is because on the one hand, sports sociology is a branch of sociology and has a close relationship with teacher education On the other hand, the clustering coefficient of nodes in the network of "Sports Sociology" is 0.6075, and this course occupies a more important position in sports education. If students majoring in sports training want to be qualified for physical education teaching, extracurricular sports training and competition work, and can engage in sports scientific research and school sports management, they need to focus on supplementary learning of the course knowledge.

Finally, the most direct motivation of the interlaminar coefficient is to study how different majors can influence each other's learning effect through common courses. It is because of the different influence between layers and within layers that we put forward the method of layered analysis, which has certain practical significance.

5.Is there a significance of clustering, or the triangulations, in general? The existence of a strong link between Sports Anatomy and Sports Physiology is hardly surprising and as the authors themselves say, “...it can only provide microscopic and partial course information and transfer behavior”.

Clustering or triangular profiling is useful because the sample size of this paper is not very large and clustering is easier and the results are intuitive and reflect the real situation. The analysis of node degrees, shortest path lengths and node meshes in a single-layer network is supplemented. From the modeled network connection diagram and complex network indicators, it is clear that the nodes of "Anatomy of Sports" and "Psychology of Sports" in the course network belong to the core nodes of the network and are the core courses. The clustering coefficients of the "Anatomy of Sport" and "Psychology of Sport" courses in the network nodes are approximately equal to 0.3536 and 0.3537, respectively, which is significantly smaller than the average clustering coefficient C0.4517, (the clustering coefficient expresses the degree of clustering between the immediate neighbours of the nodes that is also the core course). The degree of closeness, i.e. "the likelihood that two of your friends are also friends of each other"). This is because they are at the core of the network, so the clustering coefficient will be smaller than the average clustering coefficient, as shown by the node degree (the degree of a node is the number of edges connected to that node, with a node degree of 6 for both Sports Anatomy and Sports Psychology), which at threshold K = 0.4 is connected to all other courses, and the non-core courses they are connected to are not able to construct a connected edge with other courses relationships, which is highlighting the degree of centrality of the two courses. The clustering results show that clustering occurs in each discipline as well as in the other disciplines but to different degrees, which paves the way for the study of complex network models below.

6.The authors should either provide reasons for choosing the correlation threshold as 0.4, for instance - a threshold where one would get a spanning tree; or demonstrate the variation/sensitivity of the other quantities on this threshold.

We set the threshold so that the single-layer and multi-layer networks are appropriately sized. In measuring correlation, the threshold is similar to a classification tree in statistical analysis, so we find this threshold in the process of connecting edges, so that the edges and node degrees have a certain size, neither too much nor too little, so that the connections between courses can be clearly observed. If the threshold value is too small, a large number of ubiquitous nodes will be formed and the meaning of building a network will be lost; if the threshold value is too large, a fully connected network will be formed and there will be no way to characterize the composition of the network between nodes. So both values are not conducive to studying the problem from a network perspective. By taking the thresholds step by step, we found that the thresholds have a clear divide between 0.4 and 0.5, so we finally chose the relevant threshold value as 0.4.

7.I suggest that the authors look at the following additional analyses to elucidate and study the robustness of the current findings, given the fact that the sample-set is heterogeneous: (a) Divide the dataset into two (or more) non-overlapping year-spans (within 2009-14) and construct the corresponding networks; (b) Divide each of the 202 and 198 samples into two (or more) percentile ranges and perform the analysis. That is, compare the networks while considering students with either above or below average scores. In the latter case, comparing the gender aspects (male/female students) might be useful.

First of all, I would like to thank you for your valuable suggestions

(a) The data set can be divided into two parts: 2009-2011 and 2012-2014, and two complex network charts are constructed to compare and analyze the curriculum network link diagrams in different years.

(b) Calculating the average scores of students in 7 courses of two majors (sports training: 202 samples, physical education: 198 samples), sorting out the sample size of students' scores higher or lower than their average scores in 7 courses of the two majors, and taking the intersection of the scores above average and below average respectively, we can get the results that the sample sizes of 7 courses in the two majors are obtained with higher than average score and average score. Then the complex network diagram is constructed with the sample sizes of the two majors with higher than average score. The multiple network of learning transfer is constructed by coupling the same course with the course networks of different majors. We can also further analyze the gender differences of the two majors, construct a complex network map of the same courses in different majors, and observe the differences of learning transfer between male and female students.

Secondly, since the original data do not have corresponding year and student gender, we will add additional data to make our results more convincing. We have made the following changes to the valuable suggestions put forward by the experts and teachers:

We first obtained and processed the new data, and used Python software to crawl all the students' scores (from 2007 to 2019) under the public database of East China Normal University, and obtained a total of 78327 data, including: student ID, students’ name, students’ gender, major, course, course type, year, student score, etc. Then we screen the data of this score sheet, and finally determine to select the results of sports training and physical education major students from 2009 to 2019, and then screen and process the selected data, and finally get 218 sample sizes of students' academic records of sports training major and 244 sample sizes of students' academic records of physical education major.

After the completion of data acquisition and processing, in addition to the analysis of the original single-layer network chart and multi-layer network diagram, the analysis of the year, the above or below average score and gender should be carried out in the article. In terms of year analysis, considering the different courses taken by students in each academic year, the sample size between each group of years, and the problem that the number of courses will decrease with the number of years grouped, we divide the years into two groups: 2009-2016 and 2017-2019. By observing the course link network diagram of the two groups of years, we can compare the node degree, shortest path length and node betweenness in different years.

 It is more intuitive to find the core courses in the research samples. In the analysis of data with higher or lower average scores and different genders of the two majors, the analysis process we use is carried out by referring to the above (b), which is helpful to develop personalized teaching strategies for students with different scores and genders, so as to build a more perfect teaching system. On the basis of the original diagram, the course link network diagram in Figure 5-6, the link network diagram in Figure 8-11 of two sports majors in 8 courses, the correlation analysis table in Table 7-14 such as node degree statistics, shortest path length, and the second small eigenvalue table in Table 17-20 are added.

Reviewer #2:

1.The abstract seem a bit confusing, with repetition of terms.Please improve the readability of it. Results are not even mentioned or highlighted.

According to the suggestions of the expert teachers, we have rewritten the abstract section to improve the readability of the article and reduce the difficulty caused by the terminology to the reading of the article, in order to facilitate the readers to better read and understand.

Revision ideas: we have streamlined the content from the introduction section of the methods, made the language more general and focused, and made the research ideas more clear and explicitly mentioned the results.

Revision content: learning transfer is widely present in the learning of all kinds of knowledge, skills and social norms, is one of the important phenomena of learning, the reasonable use of transfer is conducive to improving the learning effect of students as well as the quality of teaching. This study starts from the data of college students' academic performance, takes real students' academic performance as a sample, measures the relevance of courses through students' academic performance, constructs various networks of learning transfer, and studies the topological structure and evolution of the networks, so as to clarify the essential law of learning transfer and make suggestions for the optimization of teaching strategies. Finally, using complex network analysis to analyze and mine the data on college students' academic performance, the article quantifies the overall structure of the courses and their hidden connections in a global and dynamic manner, and discovers the inheritance relationship between the courses, the clustering characteristics and the basic pattern of learning transfer. It also provides a platform for exploring the differences in the course structure of different majors and the learning transfer of male and female students.

2.Related works section should be improved. There are some applications including cognitive networks, knowledge acquisition and education. There are some related works that have not been mention (see e.g.doi: 10.1209/0295-5075/113/28007 doi: 10.1016/J.INS.2017.08.091and doi: 10.1038/s41598-018-20730-5).

Thank you to the reviewers for the relevant literature, we have read and studied carefully, based on which we have added the following content to enrich the original text in the related research review section. Henrique F. de Arruda et al [16][17] use complex networks for text classification, supervised classification, using a network model describing the local topology/dynamic properties of virtual words aimed at enriching the document from the imaginary differentiating information. And in 2017 research using complex network knowledge acquisition, the construction of a multi-intelligent stochastic wandering model proposed that most of the dynamics parameters have little impact on the knowledge acquisition process, and at the local scale, the choice of parameters controlling the dynamics has little impact on the performance of the considered knowledge network. Massimo Stella et al [18] used complex network research to find that the core structure of the heart vocabulary in The importance of integrating multi-relational word-word interactions in a psycholinguistic framework.

And the above literature has been added to the text.

3.Figure quality should be improved. Most of them are very hard to read given the low resolution.

Thank you very much for the correction for the problem of low-resolution graphics. We use software such as Matlab to redraw, improve the quality of the graphics, and add the corresponding text to facilitate better understanding of the reader. We hope that the expert teachers will look again, and if there are still inadequacies, we would appreciate your criticism and correction.

Due to the re-acquisition of new data, the original charts in the article are replaced by new charts, using Matlab, SPSS, Excel software to re-acquire charts, the article is supplemented and modified as follows.

(1)Use Matlab software to get the course network link diagram of Figure 4-Fig. 11 of the thesis in the format of FIG. The resolution of the graphics is 1000dpi. The height and width of the subgraph in Fig. 4 are 5-6cm and 4-5cm; those in Fig. 5-6 are 6-8cm in height and width; those in Fig. 7-11 are 7-8cm in height and 14-15cm in width.

(2) Excel is used to obtain tables 1-20, including the average score of major, variance, node degree statistics, the shortest path length between any two courses, the betweenness of each course node, the clustering coefficient of each course node, and the second smallest eigenvalue of the migration network from major 1 to specialty 2. Figure 1 is obtained to show the comparison of average scores of two majors.

(3) SPSS is used to obtain the normal distribution test chart of the course in Fig. 2-fig. 3 in the paper, and the format is EPS. The height of the graph is about 5cm and the width is 7-8cm.

4.Text should be improved. There are many errors that are definetily not acceptable. For example: "Figure3". There are no space between punctuation marks and words, and so on.

First, thanks to your correction, we carefully read the original text and corrected punctuation and other errors to increase the scientific rigor of the article. Secondly, we have checked the text several times and reorganized some of the text to improve the accuracy of the words used in the article, to make the language presentation of the article more refined, and to enhance the readability and logic of the article content.

5.Why decimal places are important? In the main manuscript the authors provided 5 decimal places. Is this really necessary?

The reviewer's considerations are indeed valid, and we have changed the data to four decimal places to facilitate comparison and analysis of the results of the data calculations. In order to make the results more rigorous and accurate, we have separated the data by year and gender, resulting in a certain loss and reduction of the sample size. Therefore, the small amount of data in the original article and the similar fluctuations between student achievement data make it impossible to reduce the number of decimal places, and setting four decimal places improves precision, which plays a crucial role in deriving experimental conclusions.

6.Results should be better discussed. What is the implication of a node having a high degree? How about assortativity.

We reconstructed the article and combed the discussion to make the results more well-argued. The degree of a node is the number of edges associated with that node, and the higher the degree, the higher the importance of the node in the network. We define the height node as the relative value of the node degree size compared under the same threshold. And at higher thresholds, the effective number of connected variables decreases and the node degrees of different courses fluctuate significantly under the division of the higher node degrees. Therefore, we need to choose a suitable threshold value to analyze and classify the node size. For example, in the construction of the network with a sample size of 462, first of all, in the case of the first row of node-degree statistics in Table 3 with a threshold value of 0.5, the average node-degree of the seven disciplines is 2, and the Sports Anatomy, Sports Psychology, and Sports Physiology with node-degree higher than 2 are classified as the first tier (which is what you call the high nodes). The next calculation with a threshold of 0.4 has a mean node degree of 5, and four disciplines with a node degree above the mean: Sports Anatomy, Sports Psychology, Sports Physiology, and Sports Statistics. However, at a threshold value of 0.5, Sports Statistics has a node degree of 1, which is much lower than the mean, and is the last course in the core, which contradicts the result. Again at a threshold of 0.4, the node degree has only two values, 5 and 6, and is not comparable. Finally, we conclude that the nodes for Sports Anatomy, Sports Psychology, and Sports Physiology are highly nodal. This is the whole process of our classification, and we hope that you will give valuable comments.

Once again, thank you very much for your constructive comments and suggestions which would help us both in English and in depth to improve the quality of the paper.

---

## [Decision Letter · Decision Letter 1]

24 Nov 2020

PONE-D-20-07521R1

An approach to the Permeation Mechanism of Learning Transfer and Teaching Strategy in Physical Education Based on Complex Network

PLOS ONE

Dear Dr. Zhao,

Thank you for submitting your manuscript to PLOS ONE. After careful consideration, we feel that it has merit but does not fully meet PLOS ONE’s publication criteria as it currently stands. Therefore, we invite you to submit a revised version of the manuscript that addresses the points raised during the review process.

We look forward to receiving your revised manuscript.

Kind regards,

Carlos Gracia-Lázaro

Academic Editor

PLOS ONE

Reviewers' comments:

Reviewer's Responses to Questions

**Comments to the Author**

1. If the authors have adequately addressed your comments raised in a previous round of review and you feel that this manuscript is now acceptable for publication, you may indicate that here to bypass the “Comments to the Author” section, enter your conflict of interest statement in the “Confidential to Editor” section, and submit your "Accept" recommendation.

Reviewer #1: All comments have been addressed

Reviewer #2: All comments have been addressed

2. Is the manuscript technically sound, and do the data support the conclusions?

Reviewer #1: Yes

Reviewer #2: Yes

3. Has the statistical analysis been performed appropriately and rigorously? 

Reviewer #1: Yes

Reviewer #2: N/A

4. Have the authors made all data underlying the findings in their manuscript fully available?

Reviewer #1: No

Reviewer #2: Yes

5. Is the manuscript presented in an intelligible fashion and written in standard English?

Reviewer #1: Yes

Reviewer #2: Yes

6. Review Comments to the Author

Reviewer #1: I recommend publication. Authors should look into the following:

1. Formatting issues, especially with references.

2. Unnecessarily having a large number of decimal places, when there is no idea about precision and error. (This was also a comment from the Reviewer 2)

3. Please include the details in the figure captions. Otherwise some captions appear to be repeated. Have a look at other articles from Plos.

4. Permeation (appears in the title) would have been a better word than infiltration. Authors should rethink.

Reviewer #2: All issues have been addressed. The authors performed an extensive analysis and now I am satisfied with the current version.

7. PLOS authors have the option to publish the peer review history of their article (what does this mean?). If published, this will include your full peer review and any attached files.

Reviewer #1: No

Reviewer #2: No

---

## [Author Response · Author response to Decision Letter 1]

30 Nov 2020

Dear Dr.Carlos (Academic Editor)and reviewers:

We appreciate you very much for your constructive comments and suggestions on our manuscript entitled”An approach to the Permeation Mechanism of Learning Transfer and Teaching Strategy in Physical Education Based on Complex Network”. We have studied all of comments carefully. According to the suggestions, we make the following explanations on the explanation and modification of the relevant contents of the paper.

1.Formatting issues, especially with references.

Thank you for your valuable suggestions. We carefully browse the articles and refer to the relevant formats of other articles in PLOS One. We have made a corrigendum to 22 references in this paper, and the relevant information of each reference has been consulted and corrected item by item The parts with problems have been corrected according to the format requirements of PLOS One, and all of them have been corrected in the paper.

2.Unnecessarily having a large number of decimal places, when there is no idea about precision and error. 

Thank you for your valuable comments. To solve this problem, we searched the PLOS One official website for papers with similar topics and found that most of the effective digits are two digits after the decimal point. We carefully refer to the opinions of the evaluation, and now we change the values in the tables such as “The shortest path length between any two courses” and “The clustering coefficient of each course node” from “four decimal places” to “two decimal places”. Because there are too many decimal places and small values in the table of “The second smallest eigenvalue of the transfer network from major 1 to 2”, if the values are kept to two decimal places, no obvious difference can be found. Therefore, the numerical value is changed into the form of scientific counting method. For example, the second small eigenvalue of “sports physiology” is changed from “0.1699” to “1.70e-1” in Table 16, which can more accurately reflect the difference between the values. I hope you will check again. If there are still some deficiencies, please criticize and correct them.

3.Please include the details in the figure captions. Otherwise some captions appear to be repeated. Have a look at other articles from Plos.

First of all, I would like to thank you for your valuable suggestions. For the suggestions given, we refer to other articles of PLOS One. In order to make the graphic title more clear, the following changes are made to some graphic titles:

(1) Single layer network diagram (page 9-18)

We changed the title of Figure 4 from "The linked network of 7 courses for two sports majors" to "The linked network of 7 courses of sports major from 2009 to 2019", and the title of Figure 5 was changed from "The course link network diagram" to "The linked network of 7courses of sports major from 2009 to 2016". The title of Figure 6 is changed from "The Course link network diagram" to "The linked network of 7courses of sports major from 2017 to 2019".

(2) Multi layer network diagram part (article page 21-29)

We changed the title of Figure 8 from " Link network of eight courses for two physical education majors" to "The linked network of 8 courses with higher than average scores of students majoring in Physical Education". The title of Figure 9 was changed from "Link network of eight courses for two physical education majors" to "The linked network of 8 courses with lower than average scores of students majoring in Physical Education", and the title of Figure 10 was changed from "Link network of eight courses for two physical education majors" to “The linked network of the scores of 8 courses for male students majoring in Physical Education”.

The title of Figure 11 is changed from "Link network of eight courses for two physical education majors" to "The linked network of the scores of 8 courses for female students majoring in Physical Education".

In addition, in order to make the graphic information display more comprehensive, we also make a more detailed description of the selection of graphics data and related information. For example:

(1) The description content of Figure 4 is “Figure 4 shows that after calculating the correlation of seven professional basic courses of 462 students majoring in sports training and physical education in East China Normal University, the different course linked network diagrams are got through dynamically changing the threshold K. The seven courses are respectively sports physiology, sports anatomy, sports statistics, sports psychology, sports introduction,Sports Sociology, Sports Sociology. The threshold value of figure a is 0.3, that of figure b is 0.4, and that of figure c is 0.5. ”

(2) The description content of Figure 7 is "Figure 7 is a multiple network of learning transfer for sports training and physical education majors of East China Normal University through the same curriculum coupled with different professional curriculum networks. Among them, there are 7 courses of the same two majors, which are sports physiology, sports anatomy, sports statistics, sports psychology, sports introduction, sports sociology, sports research methods, and different courses are "Sports Training Major - Sports Training" and "Physical Education Major - sports biochemistry". 218 samples of sports training and 244 samples of physical education were selected.” Please see the paper for other details.

4.Permeation (appears in the title) would have been a better word than infiltration. Authors should rethink.

Thank you for your suggestions. We read the article carefully and changed the term “infiltration” to “permeation”.In addition, due to the change of the author's job, the affiliated institution of the author in the article has been modified.

Finally, we would like to express our sincere thanks to all of you for your carefulness and seriousness!We look forward to receiving your feedback as soon as possible.

Kind regards,

Zhao Yi

Fudan University

---

## [Editor Report · Decision Letter 2]

1 Dec 2020

An approach to the Permeation Mechanism of Learning Transfer and Teaching Strategy in Physical Education Based on Complex Network

PONE-D-20-07521R2

Dear Dr. Zhao,

We’re pleased to inform you that your manuscript has been judged scientifically suitable for publication and will be formally accepted for publication once it meets all outstanding technical requirements.

Kind regards,

Carlos Gracia-Lázaro

Academic Editor

PLOS ONE
---

## [Editor Report · Acceptance letter]

22 Dec 2020

PONE-D-20-07521R2 

An approach to the Permeation Mechanism of Learning Transfer and Teaching Strategy in Physical Education Based on Complex Network 

Dear Dr. Zhao:

I'm pleased to inform you that your manuscript has been deemed suitable for publication in PLOS ONE. Congratulations! Your manuscript is now with our production department. 

Kind regards, 

on behalf of

Dr. Carlos Gracia-Lázaro 

Academic Editor

PLOS ONE